# From Pixels to Views: Learning Angular-Aware and Physics-Consistent Representations for Light Field Microscopy

**Feng He, Guodong Tan, Qiankun Li\*, Jun Yu\*, Quan Wen\***
*University of Science and Technology of China*
hefengcs@gmail.com, tagodong@mail.ustc.edu.cn, qklee@mail.ustc.edu.cn
harryjun@ustc.edu.cn, qwen@ustc.edu.cn
\*Corresponding Author

## Abstract

Light field microscopy (LFM) has become an emerging tool in neuroscience for large-scale neural imaging in vivo, with XLFM (eXtended Light Field Microscopy) notable for its single-exposure volumetric imaging, broad field of view, and high temporal resolution. However, learning-based 3D reconstruction in XLFM remains underdeveloped due to two core challenges: the absence of standardized datasets and the lack of methods that can efficiently model its angular–spatial structure while remaining physically grounded. We address these challenges by introducing three key contributions. First, we construct the XLFM-Zebrafish benchmark, a large-scale dataset and evaluation suite for XLFM reconstruction. Second, we propose Masked View Modeling for Light Fields (MVM-LF), a self-supervised task that learns angular priors by predicting occluded views, improving data efficiency. Third, we formulate the Optical Rendering Consistency Loss (ORC Loss), a differentiable rendering constraint that enforces alignment between predicted volumes and their PSF-based forward projections. On the XLFM-Zebrafish benchmark, our method improves PSNR by 7.7% over state-of-the-art baselines. Code and datasets are publicly available at: https://github.com/hefengcs/XLFM-Former.

## 1 Introduction

Light Field Microscopy (LFM) has emerged as a crucial technique for rapid volumetric imaging of nervous systems [18, 36, 19]. Notably, eXtended Light Field Microscopy (XLFM) [7], due to its graceful balance between speed, scale and resolution, is considered one of the most suitable LFM techniques for large-scale neural activity recording in several model organisms, including fish and mouse [1]. XLFM offers several advantages: 1)XLFM enables single-exposure acquisition of complete light field information at 100 Hz, whereas conventional microscopy techniques (e.g., two-photon microscopy [10], light-sheet microscopy [16]) require sequential layer-by-layer scanning, making it challenging to capture sub-second large-scale neural dynamics simultaneously. 2) The XLFM system incorporates a point spread function (PSF) that is approximately spatially invariant. Consequently, the reconstruction of volumes through 3D deconvolution is free from artifacts. 3) The rapid speed of XLFM allows real-time observation of large-scale population neural activity in vivo. Integrating volumetric imaging with optogenetic manipulation [3, 35] will enable optical brain-machine interface, namely closed-loop optical interrogation of brain-wide activity in both immobilized [28] and freely behaving animals [7, 4].

Despite the promise of eXtended Light Field Microscopy (XLFM) for rapid volumetric neural imaging, progress in learning-based 3D reconstruction remains limited not only due to the unique physics of XLFM, but also because of a lack of standardized datasets and evaluation protocols. First,

XLFM data differs fundamentally from conventional image data: each frame encodes a dense angular sampling of the 3D scene via a microlens array, creating highly entangled multi-view observations. Traditional convolutional models struggle to model these angular correlations and view-dependent cues effectively. In addition, raw XLFM acquisitions are abundant, producing high-quality volumetric ground truth (e.g., via Richardson–Lucy deconvolution [24]) is computationally expensive. This makes supervised learning pipelines costly at scale. Moreover, there is currently no public benchmark dataset or reproducible evaluation protocol for XLFM reconstruction. As a result, comparisons across methods are often anecdotal, and progress in the field is fragmented. Finally, most existing approaches overlook the wave-optical nature of the XLFM forward model. Without incorporating physics-guided constraints such as point spread function (PSF) priors, reconstructions may be visually plausible but physically inconsistent.

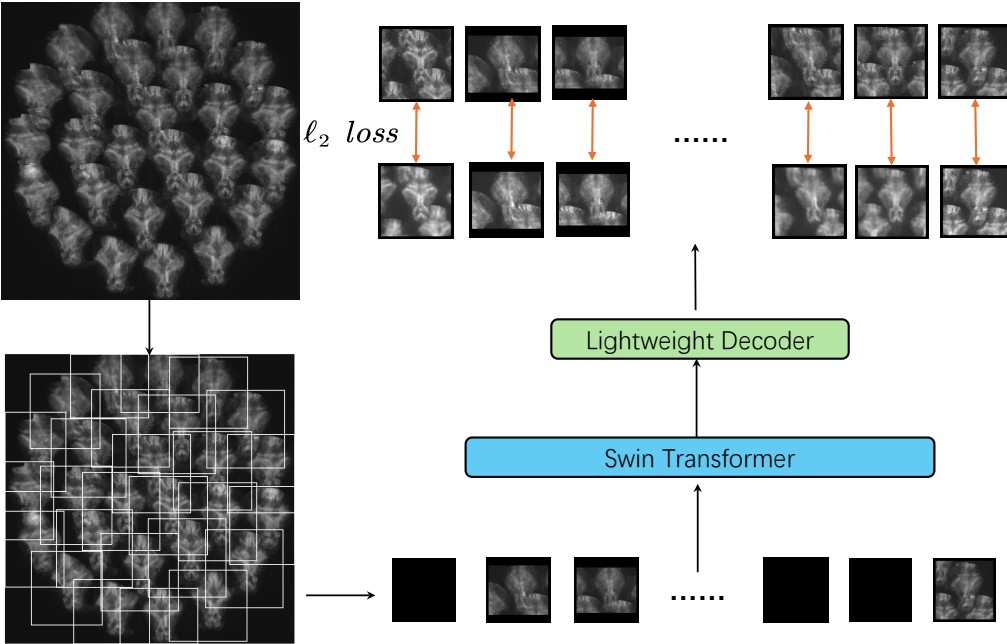

Figure 1: Our pretraining pipeline for XLFM. The raw light field acquired from the microscope is separated into 27 distinct viewpoints based on physical coordinates. With a 70% probability, we randomly mask a subset of these viewpoints and task the model with reconstructing them. The training is supervised by an $\ell_2$ loss comparing the predicted and ground-truth views.

To address the unique challenges of XLFM reconstruction, we revisit the problem not merely as a supervised regression task, but as a structured prediction problem grounded in physical optics, spatial geometry, and data asymmetry. Our design of XLFM-Former is driven by four key insights: **1) XLFM reconstruction inherently requires long-range dependency modeling across a large volumetric field-of-view with densely entangled angular observations.** While increasing the receptive field in convolutional models like U-Net can improve performance, it also incurs steep memory overhead, scaling linearly with spatial resolution and depth. Alternative global modeling strategies such as Fourier neural operators introduce even higher complexity, often transforming real-valued tensors into complex-valued representations that exceed the memory capacity of a single 80GB GPU. In contrast, we adopt a Swin Transformer encoder with hierarchical windowed attention, which efficiently captures both local and global dependencies with significantly reduced memory costs that making it a natural fit for large-scale 3D light field modeling. **2) Unlike conventional image data, XLFM views are not independent.** They exhibit occlusion patterns, spatial redundancy, and angular continuity, much like dependencies observed in natural language or multiview stereo systems. Modeling these view-wise interactions is essential for resolving fine 3D structures from ambiguous projections. We argue that the view, not the pixel, should be treated as the atomic modeling unit. Our MVM-LF pretraining task (Figure 1) reconstructs masked views from their angular neighbors, allowing the model to internalize structural priors specific to the XLFM sampling pattern.

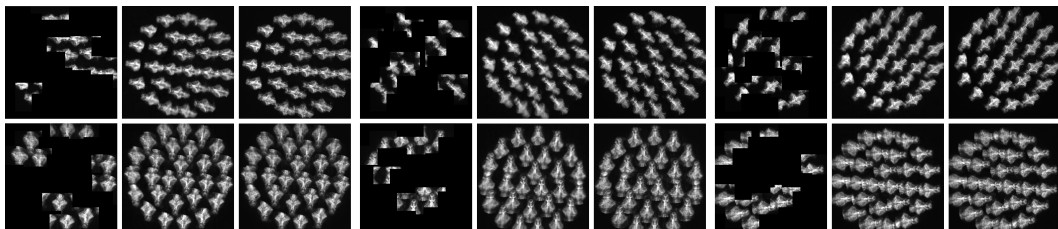

Figure 2: **The multi-view images used for pretraining an encoder model.** For each triplet, we show the masked image (left), our MVM-LF regenerated image (middle), and the ground-truth (right). The masked regions are generated by applying the binary mask complement to the original image.

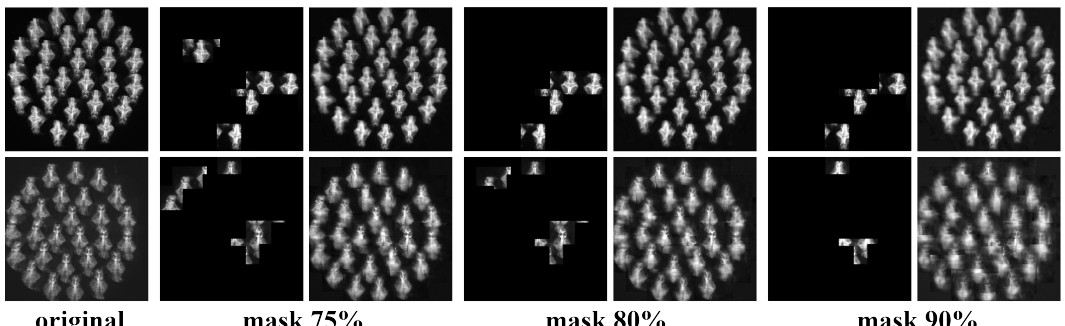

**original**          **mask 75%**          **mask 80%**          **mask 90%**

Figure 3: **Regeneration of XLFM light field images via MVM-LF.** The model can still accurately predict the view under appropriate occlusion, indicating that it has learned the global view relationship. Excessive occlusion (90%) causes prediction to crash, indicating that MVM-LF requires a reasonable occlusion ratio to balance information loss and network learning ability.

**3) The data economics of XLFM present a natural motivation for self-supervised learning.** Capturing light field images is fast, inexpensive, and non-destructive, but generating high-quality volumetric annotations (e.g., via Richardson–Lucy deconvolution) is computationally intensive. Our pretraining strategy allows the model to scale with data availability while reducing reliance on expensive labels, improving both generalization and transferability to new imaging settings. Our pretraining strategy masks a random subset of angular views and tasks the model with reconstructing them. This view-level self-supervision enables the model to scale with unlabeled XLFM data while improving generalization and transferability to unseen imaging conditions. Figure 2 illustrates the model's ability to reconstruct masked views across diverse zebrafish samples, while Figure 3 shows its robustness and limitations under varying occlusion levels. **4) Even visually accurate reconstructions may diverge from the underlying physics of light propagation.** Models trained purely on pixel-level losses often hallucinate plausible but optically inconsistent structures undermining the scientific validity of the output. To enforce physical plausibility, we introduce the Physically Optical Rendering Consistency Loss (ORC Loss), which ensures that reconstructed volumes, when passed through the known point spread function (PSF) of the imaging system, yield a synthetic light field that matches the observed measurements. This alignment with the optical forward model constrains the network to produce predictions that are both data-aligned and physics-consistent.

Our main contributions are summarized as follows:

① **A standardized benchmark for XLFM reconstruction.** We construct the first large-scale and standardized XLFM dataset, comprising 22,581 light field images captured under varying acquisition rates (10 fps / 1 fps) across three free-swimming zebrafish, seven immobilized zebrafish, and six unseen test samples. This benchmark enables reproducible evaluation and systematic advancement of XLFM reconstruction methods.

② **A transformer-based framework tailored for light field microscopy.** We develop XLFM-Former, a hierarchical Swin Transformer backbone adapted to the structural characteristics of XLFM, capable of modeling both spatial and angular dependencies efficiently across large volumetric fields.

③ **A view-masked pretraining strategy aligned with angular geometry.** We introduce Masked View Modeling for Light Fields (MVM-LF) that a self-supervised task that masks and reconstructs angular viewpoints rather than pixels. To our knowledge, this is the first pretraining strategy explicitly designed to capture inter-view structure in XLFM data, reducing dependence on costly volumetric labels.

④ **A physically grounded loss via differentiable rendering.** We formulate the Optical Rendering Consistency Loss (ORC Loss), which enforces that reconstructed 3D volumes, when forward-projected through the microscope's point spread function (PSF), match the observed light field. This loss imposes wave-optical constraints on learning, improving physical plausibility and cross-sample generalization.

## 2 Related Work

### 2.1 Unsupervised Pretraining Methods for Light Field Microscopy

In the computer vision community, unsupervised pretraining methods have gained widespread attention [12, 13, 21, 25, 2]. For example, Masked Autoencoders [14] learn by randomly masking parts of the input to help the model understand spatial relationships and global context. Contrastive learning [5, 6, 30, 17, 33, 8] creates positive and negative pairs of samples to bring similar samples closer and push dissimilar ones apart. However, in the context of LFM, unsupervised methods are still underexplored. A recent approach [34], Masked LF Modeling (MLFM), introduces a self-supervised pre-training scheme to enhance Light Field Super-Resolution (LFSSR). This method uses a transformer-based structure, XLFM-Former, to learn inter-view correlations. While this approach significantly improves performance, its reliance on random masking does not fully capture the interdependencies between views, which are critical for high-quality super-resolution. Although both approaches are applied to light fields, our proposed method is fundamentally different. First, we focus on the specific task of XLFM reconstruction. Second, our approach is based on view reconstruction, whereas theirs relies on random pixel masking.

### 2.2 3D Reconstruction in XLFM

Recent work [26, 31, 9] has demonstrated the potential of deep learning in addressing computational bottlenecks in XLFM. A recent approach [31] combines two neural networks, SLNet and XLFMNet, for real-time sparse 3D volumetric reconstruction in light field microscopy. SLNet extracts the spatio-temporally sparse components from image sequences, while XLFMNet performs high-fidelity 3D reconstruction. Another recent approach [26] proposes using a conditional normalizing flow architecture for fast 3D reconstruction of neural activity in immobilized zebrafish. **However, like XLFMNet, this method remains constrained by its sparsity-driven approach, reconstructing only neural signals while disregarding complete biological morphology.** Unlike these prior methods, XLFM-Former is designed for full-volume imaging, reconstructing not only neural activity but also entire volumetric structures. By leveraging a Swin Transformer backbone for hierarchical feature extraction and MVM-LF pretraining to learn global context dependencies, XLFM-Former provides a comprehensive reconstruction of biological samples. This distinction is critical in applications where both functional (neural signals) and anatomical (morphological structures) information are necessary for deeper biological insights.

A recent end-to-end approach [9] combines differentiable simulations of optical systems with deep learning-based reconstruction networks for high-performance computational imaging. The key insight is that global information is crucial for such problems, which is achieved by using Fourier-Nets, a shallow neural network architecture based on global kernel Fourier convolution. However, mapping to the Fourier domain results in a substantial increase in memory usage, requiring multiple GPUs for large-volume reconstruction. This limits the method's scalability and applicability to more general imaging tasks. This method is particularly expensive in terms of video memory and is not suitable for XLFM reconstruction because the final output of XLFM exceeds 100 million pixels and cannot be made into patches due to system design issues.

To achieve such global information extraction: 1) we use the Swin Transformer [22] as a feature extraction module and then apply a CNN-based decoder for feature fusion. Using self-attention is more efficient than convolution mapped to the Fourier domain. 2)To force the network to understand

the dependencies between different views, we propose a proxy task. By masking 70% of the views, we force the network to reconstruct the masked views, enabling unsupervised pretraining.

## 3 XLFM-Zebrafish Dataset

### 3.1 Data Collection

To construct the XLFM-Zebrafish Dataset, we utilized an advanced XLFM system designed to capture high-resolution volumetric neural activity in zebrafish. The data collection process was carefully structured to ensure diversity in motion states, imaging conditions, and biological variability. For free-swimming zebrafish, we recorded neural activity in an unconstrained environment, allowing for the study of brain-wide dynamics during naturalistic behaviors. A real-time tracking system was employed to continuously adjust the imaging field of view, ensuring that the zebrafish remained within the microscope's focal range. Additionally, motion artifacts caused by rapid movement were mitigated through dual-color fluorescence imaging and adaptive filtering techniques. In contrast, fixed zebrafish were embedded in a stabilizing medium to facilitate high-precision 3D structural reconstruction. This setting eliminated motion-induced distortions, enabling the extraction of detailed neural architecture. The immobilized specimens were further divided into different groups for training, validation, and testing purposes, ensuring a structured dataset for benchmarking reconstruction algorithms. To capture both rapid neural dynamics and long-term activity trends, we employed multiple imaging conditions that varied in temporal resolution and sampling strategies. This approach allowed us to balance high-fidelity reconstruction with the need for extended observation periods.

### 3.2 Dataset Statistics

To construct a high-quality XLFM-Zebrafish Dataset, we collected zebrafish in different motion states and set various sampling conditions to ensure diversity and applicability. This dataset is the first standardized XLFM zebrafish 3D reconstruction dataset, designed to evaluate the performance of deep learning models in XLFM-3D reconstruction tasks. The XLFM-Zebrafish Dataset consists of two categories of zebrafish data: Free-swimming Zebrafish, includes 3 individual zebrafish, used for studying dynamic neural activity and analyzing the impact of motion blur on 3D reconstruction. Fixed Zebrafish, includes 13 individual zebrafish, suitable for high-precision 3D structural reconstruction, with 7 individuals for training and validation, and 6 for testing. Additionally, we introduced two different sampling rates:10fps (High sampling rate): Suitable for temporal neural activity modeling and high-precision light field reconstruction. 1fps (Low sampling rate): Used for long-term dynamic tracking and reconstruction stability analysis under low frame rates. The dataset comprises 22,581 light field images. The detailed statistics are presented in the Supplementary Section B.

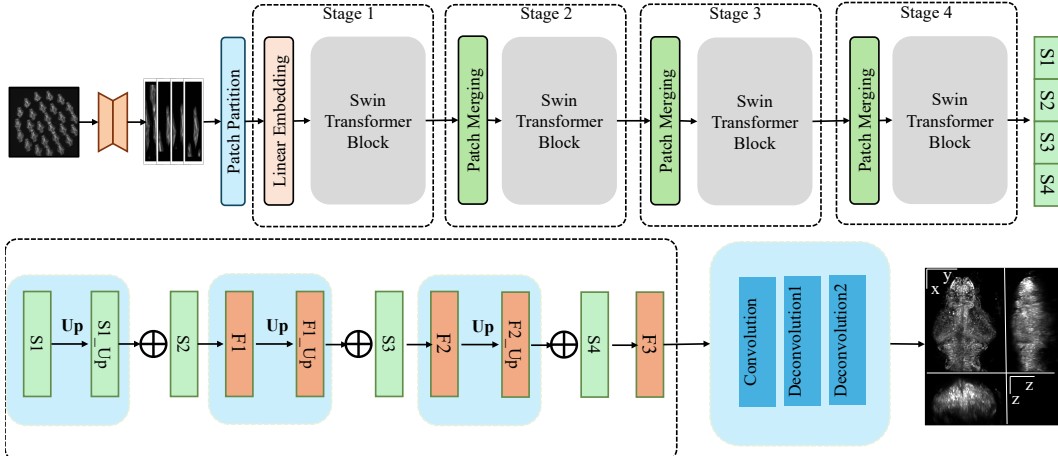

Figure 4: **Overview of the Swin-XLFM architecture.**

# 4 Methodology

The overall architecture of XLFM-Former is illustrated in Figure 4. It consists of a Swin Transformer-based encoder and a CNN decoder for progressive 3D volume reconstruction. While the encoder and decoder structures follow standard hierarchical modeling and upsampling designs, we refer the reader to Appendix A for implementation and architectural details. Here, we focus on the three core innovations: view-masked pretraining (MVM-LF), optical rendering consistency loss (ORC Loss), and the design rationale tailored to XLFM geometry.

## 4.1 Optical Rendering Consistency Loss (ORC Loss)

To ensure that reconstructed volumes not only match the ground truth structurally but also remain consistent with the underlying physical imaging process, we introduce a differentiable rendering-based supervision term: the Optical Rendering Consistency Loss (ORC Loss). This loss enforces that the predicted 3D volume, when passed through the XLFM system's forward model characterized by its Point Spread Function (PSF) produces a synthetic light field image that aligns with the observation derived from ground truth. Although one might consider comparing the reconstructed volume directly to the measured light-field image, we empirically found this approach to be highly unstable. The raw measurement contains substantial sensor noise, dark current, and scattering artifacts, which introduce non-physical gradients during training. In contrast, using the PSF-based forward projection of the ground-truth volume provides a clean and structured supervision signal, ensuring that the network learns the optical consistency without being distracted by measurement noise.

Let $\mathcal{V}_{\text{pred}}$ and $\mathcal{V}_{\text{GT}}$ denote the predicted and ground truth 3D volumes, respectively. Let $h$ be the known system-specific PSF, modeled as a 3D convolution kernel. The forward-rendered images are obtained by convolving each volume with $h$:

$$\mathbf{I}_{\text{pred}} = h * \mathcal{V}_{\text{pred}}, \quad \mathbf{I}_{\text{GT}} = h * \mathcal{V}_{\text{GT}}. \tag{1}$$

The ORC Loss is defined as the mean squared error between these two forward projections:

$$\mathcal{L}_{\text{ORC}} = \|\mathbf{I}_{\text{pred}} - \mathbf{I}_{\text{GT}}\|_2^2 = \|h * \mathcal{V}_{\text{pred}} - h * \mathcal{V}_{\text{GT}}\|_2^2. \tag{2}$$

By minimizing $\mathcal{L}_{\text{ORC}}$, the model is regularized to produce volumetric outputs that not only reconstruct anatomical structure but also render physically plausible observations under the XLFM imaging model — effectively bridging data-driven learning with wave-optical consistency.

## 4.2 Masked View Modeling for Light Fields (MVM-LF)

To enhance self-supervised learning and inter-view modeling in XLFM, we propose Masked View Modeling for Light Fields (MVM-LF) as a pretraining strategy, enabling the model to reconstruct missing views and capture global scene structures.

**Pretrained Encoder:** The encoder architecture and input representation in MVM-LF are identical to those used in XLFM reconstruction. This consistency ensures that the learned features during pretraining are directly transferable to the supervised reconstruction task. The pretrained encoder, denoted as $f_\theta$, serves as the initialization for the XLFM reconstruction model.

**Lightweight Decoder:** Inspired by self-supervised masked reconstruction frameworks, we adopt a lightweight decoder consisting of a series of convolutional layers. The decoder is responsible for predicting the missing views during pretraining. Once training is completed, the decoder is discarded, and only the pretrained encoder is retained for fine-tuning in the XLFM reconstruction task.

**Masking Strategy:** The core principle of MVM-LF is to randomly mask a proportion $r_m$ of the input views and force the network to reconstruct them based solely on the unmasked views. This proxy task compels the model to learn a joint representation of the global structure and inter-view dependencies. In practice, the masked sub-aperture views are zero-filled while their positions are preserved in the input tensor. Formally, given a set of sub-aperture views:

$$\mathcal{U} = \{U_1, U_2, \ldots, U_{N_u}\}, \tag{3}$$

we define the masked subset as:

$$\mathcal{U}_{\text{mask}} = \{U_i \mid i \in \mathcal{M}\}, \tag{4}$$

where $\mathcal{M}$ is a randomly sampled index set satisfying $|\mathcal{M}| = r_m N_u$ with $r_m = 0.7$ (i.e., 70% of the views are masked). The network is trained to reconstruct the missing views as:

$$\hat{\mathcal{U}}_{\text{mask}} = f_\theta(\mathcal{U} \setminus \mathcal{U}_{\text{mask}}). \tag{5}$$

The loss function for MVM-LF is defined as the mean squared error (MSE) between the predicted and ground truth masked views:

$$\mathcal{L}_{\text{MVM-LF}} = \sum_{U_i \in \mathcal{U}_{\text{mask}}} \|U_i - \hat{U}_i\|_2^2. \tag{6}$$

No architecture-specific changes were made during pretraining except adapting the output head, making the MVM-LF strategy applicable to any backbone capable of handling multi-view inputs.

### 4.3 Loss Function

For the pre-training task, we only use $\ell_2$ loss. For the XLFM reconstruction task, we complete the reconstruction by minimizing the following loss combination.

$$\begin{aligned}
\mathcal{L}_{\text{total}} = \frac{1}{\lambda_1}\mathcal{L}_{\text{MS\_SSIM}} + \frac{1}{\lambda_2}\mathcal{L}_{\text{Edge}} + \frac{1}{\lambda_3}\mathcal{L}_{\text{PSNR}} \\
+ \frac{1}{\lambda_4}\mathcal{L}_{\text{MSE}} + \frac{1}{\lambda_5}\mathcal{L}_{\text{ORC}}.
\end{aligned} \tag{7}$$

The detailed loss function presented in the Supplementary Section C.

## 5 Experiments

### 5.1 Implementation Details

For the MVM-LF task, we employ a batch size of 8 to facilitate stable training dynamics. To enhance convergence and mitigate the risk of the model becoming trapped in local optima, we utilize the ReduceLROnPlateau learning rate scheduler, with an initial learning rate set to 1e-4. The training process is conducted for 250 epochs to ensure robust feature learning. All experiments are performed on a distributed computing setup with four NVIDIA A100-80GB SMX4 GPUs. For the XLFM reconstruction task, the training configuration remains largely consistent, with the primary exception that the batch size is set to 1, aligning with the requirements of volumetric reconstruction. The detailed experimental setup presented in the Supplementary Section D.

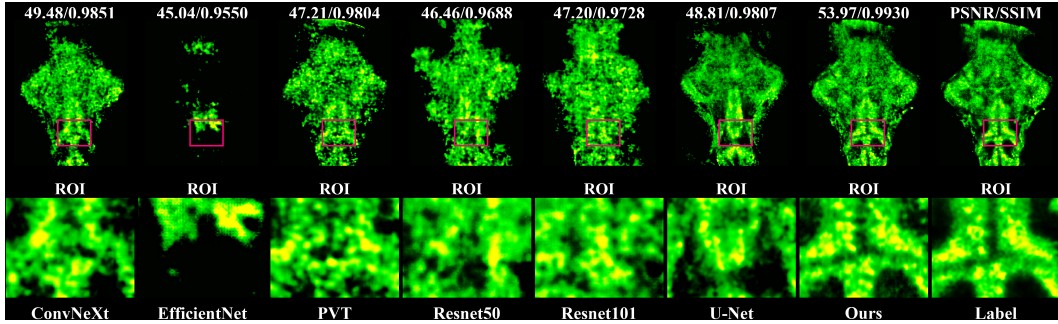

Figure 5: **Comparison with state-of-the-art architectures on the XLFM-Zebrafish Dataset.** For visualization of Zebrafish sample #1, the PSNR/SSIM values are shown in the top-left corner of each image. Additional examples on samples #2–#6 are provided in the supplementary (Figure 7).

### 5.2 Main Results

We selected the state-of-the-art architectures for comparative experiments, including ConvNeXt [23], ViT [11], PVT [32], EfficientNet [29], ResNet-50 [15], ResNet-101 [15], and U-Net [27]. As shown in Table 1, our method significantly outperforms existing state-of-the-art architectures

across all evaluation metrics. Specifically, our approach achieves the highest Peak Signal-to-Noise Ratio (PSNR) and Structural Similarity Index Measure (SSIM) on all test samples, demonstrating superior reconstruction fidelity and perceptual quality. Compared to ConvNeXt, which achieves an average PSNR of 50.16 dB and SSIM of 0.9876, our model achieves 54.04 dB and 0.9944, respectively, highlighting a substantial improvement. Similarly, our approach surpasses transformer-based models such as ViT and PVT, as well as widely used CNN architectures including EfficientNet and ResNet. Notably, U-Net, which performs competitively on some samples, is still outperformed by our model in all cases, demonstrating the effectiveness of our proposed framework. To further validate the qualitative performance of our model, Figure 5 presents visual comparisons between different methods. It is evident that our method produces reconstructions that are sharper and better aligned with the ground truth, particularly in fine structural details. In contrast, competing methods suffer from various artifacts, such as excessive blurring, structural distortions, and loss of fine details. These results indicate that our proposed approach not only achieves the best numerical performance but also generates reconstructions that are more perceptually faithful to the original structures. The integration of physics-guided constraints and transformer-based hierarchical feature modeling plays a crucial role in achieving these improvements.

Table 1: **Comparison of Methods on XLFM-Zebrafish Dataset.** The best results are highlighted in **bold**, while the second-best are underlined.

| Method | # 1 | | # 2 | | # 3 | | # 4 | | # 5 | | # 6 | | Avg. | |
|---|---|---|---|---|---|---|---|---|---|---|---|---|---|---|
| | PSNR↑ | SSIM↑ | PSNR↑ | SSIM↑ | PSNR↑ | SSIM↑ | PSNR↑ | SSIM↑ | PSNR↑ | SSIM↑ | PSNR↑ | SSIM↑ | PSNR↑ | SSIM↑ |
| ConvNeXt [23] | 49.48 | 0.9851 | 53.88 | 0.9867 | 44.87 | 0.9833 | 51.38 | 0.9882 | 51.52 | 0.9892 | 49.79 | 0.9935 | 50.16 | 0.9876 |
| ViT [11] | 49.38 | 0.9842 | 52.67 | 0.9895 | 45.29 | 0.9834 | 51.09 | 0.9888 | 51.35 | 0.9906 | 45.90 | 0.9893 | 49.28 | 0.9876 |
| PVT [32] | 47.21 | 0.9804 | 47.93 | 0.9760 | 44.50 | 0.9807 | 49.46 | 0.9851 | 48.32 | 0.9841 | 46.60 | 0.9910 | 47.34 | 0.9829 |
| EfficientNet [29] | 45.04 | 0.9550 | 54.68 | 0.9851 | 42.13 | 0.9541 | 49.56 | 0.9801 | 48.63 | 0.9772 | 27.16 | 0.7264 | 44.53 | 0.9296 |
| ResNet-50 [15] | 46.46 | 0.9688 | 54.89 | 0.9851 | 41.46 | 0.9388 | 49.47 | 0.9790 | 48.82 | 0.9786 | 39.98 | 0.9304 | 46.85 | 0.9634 |
| ResNet-101 [15] | 47.20 | 0.9728 | 54.90 | 0.9851 | 41.33 | 0.9266 | 49.47 | 0.9787 | 49.09 | 0.9800 | 39.50 | 0.8893 | 46.91 | 0.9554 |
| U-Net [27] | 48.81 | 0.9807 | 57.23 | 0.9928 | 44.41 | 0.9808 | 52.61 | 0.9908 | 52.06 | 0.9904 | 41.47 | 0.9725 | 49.43 | 0.9847 |
| Ours | **53.97** | **0.9930** | **59.83** | **0.9963** | **49.31** | **0.9910** | **54.55** | **0.9951** | **54.65** | **0.9955** | **51.95** | **0.9956** | **54.04** | **0.9944** |

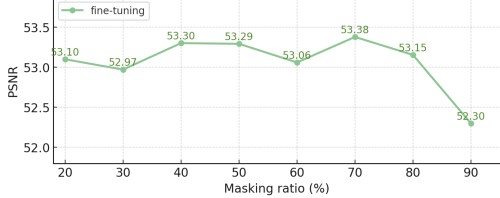

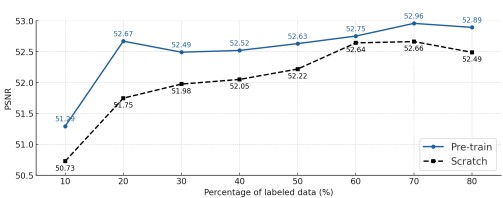

(a) Effect of masking ratio during MVM-LF pretraining.

(b) Label-efficiency (w/ vs. w/o pretraining).

Figure 6: **Effectiveness of MVM-LF pretraining.** (**Left**) PSNR under different masking ratios. (**Right**) PSNR under varying percentages of labeled data, comparing pretraining vs. scratch.

## 5.3 Ablation Study

**Masking ratio:** As shown in Figure 6a, we analyze the impact of different masking ratios in MVM-LF pretraining on XLFM reconstruction. The results indicate that moderate masking (50–70%) achieves optimal performance, with PSNR peaking at 53.38 dB at 70% masking. Lower masking ratios (20–30%) provide insufficient representation learning, leading to suboptimal fine-tuning results, while excessive masking (90%) reduces PSNR to 52.30 dB, indicating difficulty in reconstructing missing views with limited context. These findings highlight the importance of balancing information removal and reconstruction difficulty, and we adopt 70% masking as the default setting for maximal pretraining efficiency.

**Efficacy of Pre-training:** We assess data efficiency by comparing models trained from scratch and those with pretraining under varying labeled data proportions (Figure 6b). Pretraining provides a significant boost, especially in low-data regimes, with a PSNR of 51.92 dB at 10% labeled data, surpassing the 50.73 dB of training from scratch. While the performance gap narrows as more labeled data is available, the pretrained model consistently outperforms the scratch-trained counterpart, even

at 80% labeled data. These results confirm that MVM-LF pretraining enhances feature generalization, making the model more data-efficient and robust across different data availability scenarios.

**Efficacy of different components:** We conduct an ablation study to assess the impact of ORC loss and MVM-LF pretraining (Table 2). The baseline model achieves 52.14 dB PSNR, while adding ORC loss improves reconstruction fidelity to 52.96 dB. MVM-LF pretraining further enhances global view dependency learning, reaching 53.38 dB. Integrating both components into the full model yields the highest performance (54.04 dB PSNR, 0.9944 SSIM), confirming that combining physics-based constraints with self-supervised pretraining results in the most effective XLFM reconstruction.

**Efficacy of Pretraining Strategies:** We compare MVM-LF pretraining with alternative methods, including ImageNet-based initialization and pixel-level masked pretraining (Table 2). Training from scratch (Baseline) achieves 52.14 dB PSNR, while ImageNet-1k/22k pretraining provides only marginal improvements (52.70 dB / 52.38 dB), indicating that conventional pretraining is suboptimal for XLFM data. Random-masked pretraining performs slightly better (52.97 dB PSNR), but MVM-LF pretraining achieves the best results (54.04 dB PSNR, 0.9944 SSIM), demonstrating its superior ability to model multi-view dependencies. These findings highlight the importance of task-specific pretraining in optimizing XLFM reconstruction quality.

**Generalization Under Reduced Input Views**: We evaluate the pretrained model's ability to infer missing views by progressively reducing available inputs (Table 2). The scratch-trained model achieves 52.14 dB PSNR with full views, whereas the pretrained model surpasses it even with only 60% of views, reaching 52.54 dB PSNR. The best performance (53.26 dB PSNR) occurs at 80% input views, confirming strong generalization. These results demonstrate that MVM-LF pretraining enables robust multi-view reconstruction, allowing high-fidelity reconstruction even with incomplete input, making it highly adaptable to real-world imaging constraints.

Table 2: **Unified evaluation of model components, pretraining, and view-missing robustness.** PSNR/SSIM results across all configurations. Our method outperforms across all settings. The best results are highlighted in **bold**.

| Group | Setting | Notes | PSNR↑ | SSIM↑ |
|---|---|---|---|---|
| Ablation | baseline | no PSF, no MVM | 52.14 | 0.9924 |
| | + ORC loss only | w/ physics loss | 52.96 | 0.9931 |
| | + MVM-LF only | w/ view pretraining | 53.38 | 0.9938 |
| | Full (Ours) | PSF + MVM-LF | **54.04** | **0.9944** |
| Pretraining | ImageNet 1k | vision-domain weights | 52.70 | 0.9931 |
| | ImageNet 22k | large-scale weights | 52.38 | 0.9923 |
| | Random mask | pixel-masked MAE | 52.97 | 0.9934 |
| | MAE | VIT backbone | 46.55 | 0.9752 |
| | MVM-LF (Ours) | view-aware masking | **54.04** | **0.9944** |
| Missing Views | 100% (scratch) | full input, no pretrain | 52.14 | 0.9924 |
| | 90% | w/ MVM pretrain | 52.97 | 0.9933 |
| | 80% | | **53.26** | **0.9936** |
| | 70% | | 52.67 | 0.9928 |
| | 60% | | 52.54 | 0.9928 |

## 5.4 Cross-Domain Evaluation on the H2B-Nemos Dataset

We further evaluate our model on a newly collected zebrafish dataset, H2B-NeMOs, which utilizes NeMOs [20], a new genetically encoded calcium indicator, to assess cross-domain generalization across different biological conditions. This dataset involves a distinct zebrafish line and optical setup, allowing us to directly test reconstruction robustness beyond training domain.

As shown in Table 3, our XLFM-Former consistently outperforms a range of representative architectures under identical training settings with ResNet-101 as the common baseline. Notably, our model achieves a +0.92 dB PSNR gain in the supervised setting and a +2.29 dB gain in the zero-shot setting, confirming its ability to generalize across imaging domains.

Table 3: Comparison on the H2B-Nemos dataset (baseline: ResNet-101).

| Method | PSNR ↑ | SSIM ↑ | ΔPSNR vs. R101 ↑ | ΔSSIM vs. R101 ↑ |
|---|---|---|---|---|
| EfficientNet | 49.01 | 0.9826 | –2.42 | –0.0118 |
| ViT-tiny | 50.87 | 0.9904 | –0.55 | –0.0041 |
| PVT-tiny | 50.89 | 0.9903 | –0.53 | –0.0042 |
| ConvNeXt-tiny | 50.88 | 0.9903 | –0.54 | –0.0042 |
| U-Net | 51.00 | 0.9910 | –0.42 | –0.0035 |
| ResNet-50 | 51.34 | 0.9931 | –0.09 | –0.0014 |
| ResNet-101 (baseline) | 51.42 | 0.9945 | – | – |
| **XLFM-Former (Ours, Full)** | **52.34** | **0.9955** | **+0.92** | **+0.0010** |
| **XLFM-Former (Zero-Shot)** | **53.72** | 0.9930 | **+2.29** | –0.0015 |

## 5.5 Robustness to PSF Mis-Calibration

We further assess the robustness of our Optical Reconstruction Consistency (ORC) loss to inaccuracies in the forward model by intentionally perturbing the point spread function (PSF) used in the forward projection. Specifically, we vary the axial full-width at half-maximum (FWHM) of the PSF by ±10% to simulate mild miscalibration or optical aberrations. As shown in Table 4, the reconstruction performance remains highly stable under these perturbations, with PSNR fluctuations within ±0.12 dB and SSIM deviations within ±0.0002.

This stability arises because the ORC loss enforces consistency between the reconstructed and measured light-field projections at a global level, making it less sensitive to small inaccuracies in PSF calibration. Such robustness is particularly desirable for real-world microscopy setups, where slight deviations in PSF shape or system alignment are inevitable.

Table 4: Robustness of ORC loss to PSF perturbations on the H2B-Nemos dataset.

| PSF Setting | PSNR ↑ | ΔPSNR ↑ | SSIM ↑ | ΔSSIM ↑ |
|---|---|---|---|---|
| Baseline | 52.3440 | – | 0.9955 | – |
| PSF +10% FWHM | 52.3369 | –0.0071 | 0.9953 | –0.0002 |
| PSF –10% FWHM | 52.4647 | +0.1207 | 0.9957 | +0.0002 |

These results demonstrate that XLFM-Former does not rely on perfect PSF calibration, and can tolerate mild optical misalignments, reducing the need for exact calibration in practical deployments.

## 6 Conclusion

We introduce XLFM-Former, a unified learning framework that combines physics-based constraints and self-supervised angular modeling to enable high-speed, high-fidelity volumetric imaging in eXtended Light Field Microscopy (XLFM). By integrating a hierarchical transformer backbone, a differentiable rendering loss via PSF, and a view-masked pretraining strategy (MVM-LF), our method captures the geometric and optical structures inherent in light field data with minimal supervision. Empirical results demonstrate that XLFM-Former significantly improves over existing approaches in PSNR and SSIM, especially under limited labels or incomplete input views. Our ablations further reveal the complementary roles of physical priors and angular-aware self-supervision in robust 3D reconstruction. Beyond the scope of XLFM, this work highlights a scalable path toward physics-aligned, data-efficient learning for scientific imaging. We believe our findings bridge neural imaging and modern vision learning, offering a transferable foundation for future exploration of self-supervised, physically informed learning paradigms in neuroscience, biomedicine, and beyond.

**Limitations:** Despite promising results, our approach is evaluated only on zebrafish datasets, limiting generalization to other organisms such as mouse or drosophila. We focus on demonstrating the physical and computational efficiency of XLFM-Former rather than directly assessing functional trace extraction. As neural signal extraction involves complex pipelines with registration, segmentation, and clustering, future work will explore direct trace extraction to enhance biological analysis.

## Acknowledgments

We would like to thank Professor Pengcheng Zhou for his helpful discussions during the development of the PSF-guided loss function. This research was supported by the STI2030–Major Projects (Grant No. 2022ZD0211900), "Brain Science and Brain-Inspired Intelligence", under the subproject "New Technologies for Whole-Brain-Scale Neuronal Mesoscopic Atlas" (Grant No. 2022ZD0211904). The numerical calculations in this paper were performed on the supercomputing system at the Supercomputing Center of the University of Science and Technology of China.

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

## A    XLFM-Former

Figure 4 illustrates the overall architecture of XLFM-Former. We describe the details of encoder and decoder in this subsection.

**Encoder:** The input to XLFM-Former is a raw light field captured by the XLFM system. The original light field data is represented as a collection of sub-aperture channels, each encoding spatial information of the observed scene and collectively forming a multi-view representation. These sub-aperture channels correspond to distinct angular responses induced by the XLFM optical system, without assuming a regular grid of physical viewpoints as in standard MLA-based light field imaging. To avoid ambiguity, we clarify that all sub-aperture channels are concatenated along the channel axis, forming a tensor of size $H \times W \times C$. A cropping operation is applied to align the data with the optical coordinate system, yielding $\mathcal{X} \in \mathbb{R}^{H \times W \times D \times S}$, where $H$ and $W$ denote spatial dimensions, $D$ corresponds to the axial depth resolution from cropping, and $S = 1$ is the channel number in XLFM imaging. Importantly, the sub-aperture index is treated as channel information rather than a spatial dimension.

To facilitate self-attention computation in the transformer-based encoder, the patch partitioning layer divides the input into a sequence of non-overlapping 2D patches of size $(H', W')$ for each depth slice $D$. The resulting tokenized feature map has dimensions $\frac{H}{H'} \times \frac{W}{W'} \times D$, where each patch is projected into a $C$-dimensional embedding space using a linear embedding layer. We explicitly note that the encoder adopts **2D positional embedding** along the $(H, W)$ axes only; no positional encoding is assigned to the sub-aperture dimension since it is folded into channels. This patch-wise tokenization enables efficient modeling of spatial and depth information without ambiguity about angular indexing.

To effectively capture multi-scale features, XLFM-Former employs a hierarchical encoding strategy based on the Swin Transformer. The encoder consists of four stages, each with two consecutive Swin Transformer blocks. At the end of each stage, a patch merging layer downsamples the resolution while increasing the feature dimension. Given an input feature map $\mathbf{S} \in \mathbb{R}^{h \times w \times d \times c}$ at stage $T$, the

patch merging operation groups adjacent $2 \times 2$ patches in the $(h, w)$ plane (applied per depth slice) and concatenates their features, yielding $\mathbf{S}' \in \mathbb{R}^{\frac{h}{2} \times \frac{w}{2} \times d \times 4c}$. A linear projection then reduces the feature dimension to $2c$:

$$\mathbf{S}_{T+1} = \mathbf{W}\mathbf{X}' + \mathbf{b},$$

where $\mathbf{W}$ and $\mathbf{b}$ are learnable parameters.

The four hierarchical feature maps $\{S_1, S_2, S_3, S_4\}$ are passed to the decoder via skip connections. Each decoder stage upsamples the features and fuses them with the corresponding encoder outputs, ensuring that both high-level semantics and fine-grained spatial details are preserved.

**Decoder**: The decoder reconstructs the final high-resolution 3D structure by progressively integrating multi-scale features extracted from the encoder. This process consists of two key components: progressive upsampling and cross-level feature fusion. The decoder takes the hierarchical feature maps from the encoder, denoted as $S_i, \quad i \in \{1, 2, 3\}$, and reconstructs the high-resolution output through a sequence of deconvolutional operations. The highest-level feature map $S_1$ serves as the starting point, and at each stage, an upsampling operation is applied to gradually recover spatial details. Fused feature maps $F_i, \quad i \in \{1, 2, 3\}$ are obtained by adding features of different scales:

$$\hat{F}_0 = S_1, \quad \hat{F}_i = \text{Up}(\hat{F}_{i-1}) + S_i, \quad i \in \{1, 2, 3\}. \tag{8}$$

The upsampling is performed using deconvolution layers:

$$\hat{S}_i = \text{Deconv}_{i+1 \to i}(\hat{S}_{i+1}), \quad i \in \{1, 2, 3\}, \quad \hat{S}_4 = S_4. \tag{9}$$

$$\hat{S}_4 = S_4, \quad \hat{S}_i = \text{Deconv}_{i+1 \to i}(\hat{S}_{i+1}), \quad i \in \{1, 2, 3\}. \tag{10}$$

To enhance reconstruction quality, cross-level feature fusion is applied at each stage, allowing fine-grained spatial information from lower-level features to be combined with high-level semantic information. The fusion is performed by summing the upsampled feature maps with the corresponding encoder feature maps, followed by a convolutional refinement:

$$F_i = \text{Conv}(S_i + \hat{S}_i), \quad i \in \{1, 2, 3\}. \tag{11}$$

The final reconstructed 3D structure is obtained by applying a reconstruction head, which maps the fused feature $F_3$ to the target 3D volume: First, a convolution is performed to match the number of channels:

$$\hat{F} = \text{Conv}_{1 \times 1}(F_{\text{fusion}}), \tag{12}$$

Then two deconvolutions are performed to match the target resolution:

$$\mathcal{V}_{\text{pred}} = \text{Conv}_{1 \times 1}\left(\text{Deconv}_2\left(\text{Deconv}_1(\hat{F})\right)\right). \tag{13}$$

$$\mathcal{V}_{\text{pred}} = \text{Deconv}_2\left(\text{Deconv}_1\left(\text{Conv}_{1 \times 1}(\hat{F})\right)\right). \tag{14}$$

This approach ensures that the final output preserves both deep semantic features and high-frequency spatial details, achieving high-quality 3D reconstruction.

XLFM-Former leverages Swin Transformer to extract multi-scale features and semantic information. The extracted multi-scale features undergo global and local feature fusion within the encoder. Figure 4 illustrates the overall framework of XLFM-Former.

**Feature Extractor:** We adopt Swin Transformer [22] as the backbone network to extract hierarchical features $F_0, F_1, F_2, F_3$, which represent multi-scale features from shallow to deep. Among them, $F_0$ has a higher spatial resolution but weaker semantic information, while $F_3$ contains strong semantic information but fewer spatial details.

$$F_{\text{encoder}} = \text{SwinTransformer}(I)$$

where $I$ is the input image, and $F_{\text{encoder}} = \{F_0, F_1, F_2, F_3\}$ represents the feature pyramid extracted by the Swin Transformer.

**Pyramid Fusion Module:** Pyramid Fusion uses Progressive Upsampling and Cross-Level Feature Fusion to achieve the interaction of multi-scale information.

The highest-level feature $F_3$ is upsampled through **Deconvolution**, then fused with $F_2$:

$$\hat{F}_2 = \text{Conv}(F_2 + \text{Deconv}_{3 \to 2}(F_3))$$

The fusion continues layer by layer until the highest resolution is restored:

$$\hat{F}_1 = \text{Conv}(F_1 + \text{Deconv}_{2\to1}(\hat{F}_2))$$

$$F_{\text{fusion}} = \text{Conv}(F_0 + \text{Deconv}_{1\to0}(\hat{F}_1))$$

The final fused feature map $F_{\text{fusion}}$ simultaneously contains deep semantic information and fine-grained details.

## B   Dataset Statistics Details

Table 5 summarizes the statistics of the XLFM-Zebrafish dataset, which consists of image data from zebrafish in both free-swimming and fixed conditions. The dataset is categorized into Pre-training Set, Training/Validation Set, and Test Set to facilitate different stages of model development. The dataset is carefully designed to ensure diversity in motion complexity, viewpoint variations, and temporal resolutions. By pretraining on large-scale, complex free-swimming zebrafish data, the model gains a stronger ability to generalize and better reconstruct simpler fixed zebrafish data, leading to improved performance. This structured dataset and training methodology provide a standardized benchmark for evaluating XLFM-Former and contribute to the advancement of XLFM-based 3D reconstruction techniques.

**Pre-training Set (Free-swimming Zebrafish):** The Pre-training Set comprises three free-swimming zebrafish (m1, m2, m3), totaling 20,123 images, all captured at a 10 fps sampling rate. Free-swimming zebrafish exhibit highly dynamic and complex motion, leading to significant variations in viewpoint and pose. This complexity presents a challenge for 3D reconstruction but also offers an opportunity for the model to learn richer geometric structures. To leverage this complexity, we employ unsupervised pretraining on this large-scale free-swimming dataset before training on the fixed zebrafish dataset. By learning from diverse, naturally occurring light field transformations, the model develops a robust understanding of light field geometry and depth relationships. This approach significantly improves performance when fine-tuned on simpler fixed zebrafish data, demonstrating the benefits of pretraining on large, diverse datasets before supervised learning on smaller, more controlled datasets.

**Training/Validation Set (Fixed Zebrafish):** The Training/Validation Set contains seven fixed zebrafish (f1-f7) with a total of 1,761 images. Most samples were collected at 10 fps, while some (e.g., f3) were acquired at 1 fps to introduce variations in temporal resolution. Compared to free-swimming zebrafish, the fixed zebrafish dataset presents a more structured and constrained setting, making it an ideal target for supervised training once the model has been pretrained on more complex free-swimming data.

**Test Set (Fixed Zebrafish):** The Test Set consists of six fixed zebrafish (t1-t6) with a total of 1,011 images. Some samples (e.g., t2, t4, t5) were acquired at 1 fps, allowing a comprehensive evaluation of XLFM-Former's reconstruction performance under different sampling conditions.

## C   Loss Function Details

In our proposed XLFM-Former framework, we employ different loss functions tailored for **pretraining** and **reconstruction tasks** to ensure robust and high-quality 3D volume generation.

**Pretraining Loss:** For the self-supervised pretraining task, where we use Masked View Modeling-Light Field (MVM-LF), we adopt a simple $\ell_2$ loss to enforce the consistency between the predicted and ground truth light field views:

$$\mathcal{L}_{\text{pretrain}} = \|\hat{I} - I\|_2^2, \tag{15}$$

where $\hat{I}$ represents the predicted light field views, and $I$ denotes the original (ground truth) views before masking. The choice of $\ell_2$ loss is motivated by its stability in regression tasks and its ability to ensure smooth reconstructions during pretraining.

**XLFM Reconstruction Loss:** For the final **3D reconstruction task**, we minimize the following composite loss function:

Table 5: **The XLFM-Zebrafish dataset statistics.**

| Dataset Name | Number of Images | Sampling Rate (fps) |
|---|---|---|
| **Free-swimming Zebrafish (Pre-training Set)** | | |
| moving_fish1 (m1) | 4000 | 10 |
| moving_fish2 (m2) | 7332 | 10 |
| moving_fish3 (m3) | 8791 | 10 |
| **Fixed Zebrafish (Training/Validation Set)** | | |
| fixed_fish1 (f1) | 240 | 10 |
| fixed_fish2 (f2) | 117 | 10 |
| fixed_fish3 (f3) | 318 | 1 |
| fixed_fish4 (f4) | 314 | 10 |
| fixed_fish5 (f5) | 374 | 10 |
| fixed_fish6 (f6) | 214 | 10 |
| fixed_fish7 (f7) | 184 | 10 |
| **Fixed Zebrafish (Test Set)** | | |
| test_fixed_fish1 (t1) | 300 | 10 |
| test_fixed_fish2 (t2) | 41 | 1 |
| test_fixed_fish3 (t3) | 334 | 10 |
| test_fixed_fish4 (t4) | 61 | 1 |
| test_fixed_fish5 (t5) | 61 | 1 |
| test_fixed_fish6 (t6) | 214 | 10 |
| H2B-Nemos | | |
| test_fixed_fish1 (t1) | 73 | 10 |
| test_fixed_fish2 (t2) | 73 | 10 |
| test_fixed_fish3 (t3) | 73 | 10 |
| test_fixed_fish4 (t4) | 73 | 10 |

$$\mathcal{L}_{\text{total}} = \frac{1}{\lambda_1}\mathcal{L}_{\text{MS\_SSIM}} + \frac{1}{\lambda_2}\mathcal{L}_{\text{Edge}} + \frac{1}{\lambda_3}\mathcal{L}_{\text{PSNR}}$$
$$+ \frac{1}{\lambda_4}\mathcal{L}_{\text{MSE}} + \frac{1}{\lambda_5}\mathcal{L}_{\text{ORC}}. \tag{16}$$

Each term in the loss function contributes to a different aspect of the 3D reconstruction quality, ensuring sharpness, accuracy, and optical consistency. Below, we provide a detailed breakdown of each component:

**Multi-Scale Structural Similarity (MS-SSIM) Loss:** Structural Similarity Index (SSIM) is widely used to measure the perceptual similarity between images. We employ a multi-scale SSIM (MS-SSIM) loss to capture both local and global structural fidelity:

$$\mathcal{L}_{\text{MS\_SSIM}} = 1 - \text{MS-SSIM}(\hat{V}, V), \tag{17}$$

where $\hat{V}$ and $V$ represent the predicted and ground truth 3D volumes. MS-SSIM helps preserve structural details and enhances the perceptual quality of the reconstruction.

**Edge-Aware Loss:** To enhance edge sharpness and suppress blurring, we define the edge-preserving loss as a weighted combination of edge loss and multi-scale MSE loss:

$$\mathcal{L}_{\text{Edge\_Aware}} = \frac{1}{\lambda_{\text{edge}}}\Big( \|\nabla_x\hat{V} - \nabla_x V\|_1 + \|\nabla_y\hat{V} - \nabla_y V\|_1$$
$$+ \|\nabla_z\hat{V} - \nabla_z V\|_1 \Big)$$
$$+ \frac{1}{\lambda_{\text{mse}}}\sum_{s=1}^{S}\|\hat{V}^{(s)} - V^{(s)}\|_2^2. \tag{18}$$

Here, $\mathcal{L}_{\text{edges\_loss}}$ ensures sharpness by penalizing gradient differences along spatial dimensions, while $\mathcal{L}_{\text{multi\_scale\_MSE}}$ enforces consistency across multiple resolutions, improving global structure and fine details.

**Peak Signal-to-Noise Ratio (PSNR) Loss:** PSNR is a widely used metric for measuring signal fidelity. We define a loss function that penalizes low PSNR values:

Table 6: **Loss function.** 'Full' denotes the complete model with all loss terms. 'w.o. ms_ssim' removes the multi-scale SSIM loss. 'w.o. Edge_Aware' omits the edge-aware loss. 'w.o. PSNR' excludes the PSNR loss. 'w.o. MSE Loss' removes the MSE loss. The best results are highlighted in **bold.**

| Method | PSNR↑ | SSIM↑ |
|---|---|---|
| w.o. $\mathcal{L}_{MS\_SSIM}$ | 53.2787 | 0.9937 |
| w.o. $\mathcal{L}_{Edge\_Aware}$ | 52.1870 | 0.9922 |
| w.o. $\mathcal{L}_{PSNR}$ | 53.0741 | 0.9935 |
| w.o. $\mathcal{L}_{MSE}$ | 53.2521 | 0.9937 |
| Full | **54.0435** | **0.9944** |

$$\mathcal{L}_{\text{PSNR}} = -\text{PSNR}(\hat{V}, V), \tag{19}$$

where a higher PSNR corresponds to higher-quality reconstructions. By minimizing this loss, we encourage the model to reduce noise and artifacts. The PSNR term is included as an auxiliary component that provides a consistent yet minor gain in reconstruction stability, rather than representing a core novelty of our approach.

**Mean Squared Error (MSE) Loss:** The **MSE loss** ensures pixel-wise intensity similarity between the predicted and ground truth volumes:

$$\mathcal{L}_{\text{MSE}} = \|\hat{V} - V\|_2^2. \tag{20}$$

While MSE is commonly used in image restoration, it is prone to blurring. Thus, we use it in combination with perceptual losses (e.g., MS-SSIM and edge loss) to balance fine details and overall similarity.

## C.1 Ablation Study on Loss Functions

To evaluate the impact of different loss components on the overall performance of our model, we conduct an ablation study by removing individual loss terms and measuring the performance in terms of PSNR and SSIM. The results are summarized in Table 6.

From the table, we observe that the complete model (*Full*) achieves the highest PSNR of **54.0435** and SSIM of **0.9944**, demonstrating the effectiveness of integrating all loss terms.

**Effect of Multi-Scale SSIM Loss:** Removing the multi-scale SSIM loss (*w.o.* $\mathcal{L}_{MS\_SSIM}$) results in a performance drop, reducing PSNR to 53.2787 and SSIM to 0.9937. This highlights the importance of SSIM-based perceptual loss in preserving structural information.

**Effect of Edge-Aware Loss:** When the edge-aware loss is removed (*w.o.* $\mathcal{L}_{Edge\_Aware}$), the PSNR decreases significantly to 52.1870, while SSIM drops to 0.9922. This indicates that the edge-aware term plays a crucial role in maintaining sharp details and edge consistency.

**Effect of PSNR Loss:** Excluding the PSNR-based loss (*w.o.* $\mathcal{L}_{PSNR}$) results in a minor degradation in performance, with PSNR reducing to 53.0741 and SSIM to 0.9935. This suggests that optimizing directly for PSNR provides some benefits but is not the dominant factor.

**Effect of MSE Loss:** When the MSE loss is removed (*w.o.* $\mathcal{L}_{MSE}$), the PSNR slightly drops to 53.2521, while SSIM remains at 0.9937. This indicates that MSE contributes to pixel-wise accuracy but is less critical than the other loss terms.

## D Training Configuration

For the MVM-LF task, we adopt a batch size of 8 to ensure stable training dynamics while maintaining an efficient balance between computational cost and convergence stability. To further enhance the training process and prevent the model from becoming trapped in local optima, we employ the ReduceLROnPlateau learning rate scheduler. The initial learning rate is set to 1e-4, and the scheduler

Table 7: Data augmentation techniques applied during training. Each transformation is applied with a probability of 0.5.

| Augmentation Type | Probability |
|---|---|
| Random Flip | 0.5 |
| Random Rotation | 0.5 |
| Random Gaussian Noise&Blur | 0.5 |
| Random Brightness &Contrast | 0.5 |

dynamically adjusts the learning rate based on validation loss fluctuations, reducing it when no improvement is observed for a predefined number of epochs. This adaptive learning rate strategy helps in maintaining a steady convergence while preventing premature stagnation in suboptimal solutions.

To achieve robust feature learning and ensure generalization across diverse data distributions, we train the model for 250 epochs. Given the complexity of the task and the high-dimensional nature of the input data, prolonged training allows the model to capture intricate spatial and structural information effectively. All experiments are conducted in a distributed computing environment equipped with four NVIDIA A100-80GB SMX4 GPUs, leveraging multi-GPU parallelism to accelerate training and optimize resource utilization.

For the XLFM reconstruction task, the training setup remains largely consistent with the MVM-LF configuration. However, a notable difference is the use of a batch size of 1, which aligns with the requirements of volumetric reconstruction. Given that volumetric data often involves higher memory footprints due to its three-dimensional representation, a smaller batch size ensures that computations remain feasible within available GPU memory constraints. All experiments are implemented using PyTorch Lightning, a high-level deep learning framework that simplifies training and enhances reproducibility.

## D.1 Data Augmentation Strategy

To improve the model's robustness and generalization capability, we apply a series of data augmentation techniques during training. These augmentations are applied with a probability of 0.5, as summarized in Table 7.

These augmentation techniques enhance the model's ability to handle variations in real-world data, reducing overfitting and improving generalization performance.

## D.2 Model Architecture

For all experiments, we adopt the Swin Transformer framework in its tiny configuration. The Swin Transformer is a hierarchical vision transformer that efficiently models long-range dependencies while maintaining computational efficiency. The tiny variant provides a lightweight architecture suitable for our training setup, balancing performance and efficiency.

## D.3 Extended Training on Additional Datasets

To further explore the scalability and generalization capability of XLFM-Former, we conducted large-scale training on an extended dataset. This additional training aimed to improve the model's ability to reconstruct fine-grained details while enhancing robustness across diverse volumetric data.

The large-scale training process required substantial computational resources, consuming approximately 1344 A100-80GB GPU hours. This extensive training allowed the model to refine its feature representation and leverage a broader data distribution, leading to improved reconstruction quality.

To demonstrate the effectiveness of this large-scale training, we provide a visualized demo showcasing the enhanced view synthesis capability. The qualitative results highlight the model's ability to generate high-fidelity reconstructions with improved structural consistency and perceptual quality, further validating the benefits of extended training.

## D.4 Evaluation Details

All reconstructed volumes are normalized within the physical intensity range of $[0, 2000]$, corresponding to the effective dynamic range of the neural fluorescence signals captured by our XLFM system. This differs from conventional RGB-based normalization ($[0, 255]$), and may lead to slightly higher absolute PSNR/SSIM values compared with typical computer vision benchmarks. To ensure fairness, all compared methods are evaluated under the same normalization setting.

# E    Qualitative Comparison on XLFM-Zebrafish Dataset

To better understand the reconstruction quality beyond numerical metrics, we present qualitative comparisons in Figure 7. For each zebrafish sample, we visualize the XY projection and a zoomed-in region-of-interest (ROI) across baseline methods, our proposed model, and the ground truth.

We observe that convolution-based methods (e.g., ConvNeXt, EfficientNet) tend to oversmooth fine neural structures or introduce artifacts. Transformers such as PVT improve spatial continuity but still suffer from blurry ROIs. In contrast, XLFM-Former accurately recovers elongated dendritic shapes and high-frequency details, closely matching the ground truth.

# F    H2B-Nemos datasets

## F.1    Cross-Dataset Ablation Study on the H2B-Nemos Dataset

To further assess the robustness and generalizability of our model design, we conduct an ablation study on the newly collected H2B-Nemos dataset, which differs from the original G8S dataset in imaging conditions and noise statistics. All architectural and loss configurations remain unchanged, except for a slight adjustment to the ORC loss weight (from $1 \times 10^{-4}$ to $3 \times 10^{-5}$) to account for minor differences in noise level.

As shown in Table 8, our approach maintains consistent improvement trends across modules and loss components. Each component contributes positively to reconstruction quality, demonstrating that the design principles established on the G8S dataset generalize well to new imaging domains.

Table 8: Cross-dataset ablation study on the H2B-Nemos dataset.

| Method | PSNR ↑ | ΔPSNR ↑ | SSIM ↑ | ΔSSIM ↑ |
|---|---|---|---|---|
| Baseline | 50.87 | – | 0.9903 | – |
| + MVM-LF only | 51.66 | +0.79 | 0.9940 | +0.0037 |
| + ORC Loss only | 51.77 | +0.90 | 0.9935 | +0.0032 |
| **Full (Ours)** | **52.34** | **+1.47** | **0.9955** | **+0.0052** |

These results confirm that the proposed components both architectural and loss-based remain effective across different zebrafish datasets, highlighting the robustness of XLFM-Former.

## F.2    Robustness to ORC Loss Weight Variation

We further examine the sensitivity of our model to the weighting coefficient of the Optical Reconstruction Consistency (ORC) loss. As shown in Table 9, the performance remains highly stable across a wide range of ORC weights, demonstrating that the method is not fragile or over-tuned to a specific configuration.

Specifically, sweeping the ORC weight from $10^{-4}$ to $10^{-2}$ results in only minor fluctuations in PSNR and SSIM. Weights that are too small (e.g., $10^{-4}$) may lead to unstable convergence, while larger values (e.g., $5 \times 10^{-3}$ to $10^{-2}$) yield consistently strong performance and stable optimization behavior. These results confirm that the ORC formulation provides reliable supervision across diverse imaging conditions without requiring fine-grained tuning.

Slight tuning is expected when transferring to new datasets due to differences in fluorescence properties or noise characteristics. Importantly, no changes to the architecture or training strategy are required, underscoring the generality and robustness of the proposed ORC formulation.

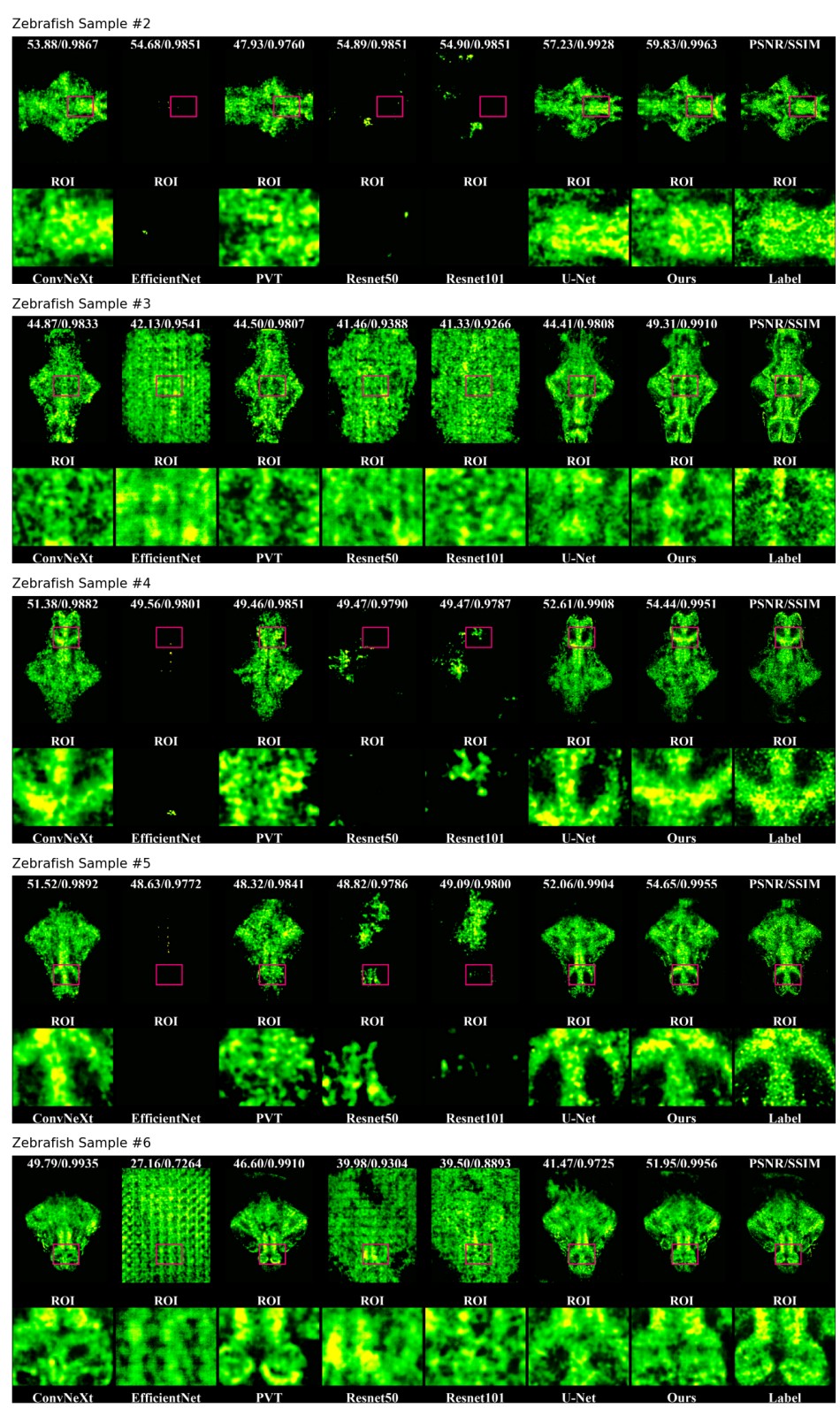

Figure 7: **Qualitative comparison on XLFM-Zebrafish samples #2–#6.** Each row shows a different zebrafish sample. For each method, the XY projection and ROI zoom-in are shown. PSNR/SSIM values are displayed at the top-left corner of each full-frame image. The visualizations confirm that our method consistently preserves neural structures across diverse samples and acquisition conditions.

Table 9: ORC loss weight sweep showing the stability of XLFM-Former under different loss configurations.

| ORC Loss Weight | PSNR ↑ | SSIM ↑ | Observation |
|---|---|---|---|
| $1 \times 10^{-4}$ | 50.98 | 0.991 | Unstable convergence |
| $1 \times 10^{-3}$ | 51.70 | 0.994 | Lower performance |
| $5 \times 10^{-3}$ | 52.34 | 0.995 | Best, stable convergence |
| $5 \times 10^{-2}$ | 52.23 | 0.995 | Slightly lower than optimal |
| $1 \times 10^{-2}$ | 52.35 | 0.995 | Similar but less stable |

## F.3 Comparison with Classical Iterative Deconvolution

We further compare our method with the classical iterative Richardson–Lucy Deconvolution (RLD) algorithm under different iteration counts (20, 30, and 40) on the challenging H2B-Nemos dataset. While RLD can achieve very high PSNR and SSIM when run for many iterations, it incurs orders of magnitude higher memory usage and significantly lower speed, making it impractical for real-time or high-throughput applications.

Table 10: Comparison with the classical Richardson–Lucy Deconvolution (RLD) on the H2B-Nemos dataset.

| Method | PSNR ↑ | SSIM ↑ | FPS ↑ | Peak Memory ↓ (MiB) |
|---|---|---|---|---|
| RLD-20 | 67.36 | 0.9987 | 0.068 | 20899 |
| RLD-30 | 69.85 | 0.9994 | 0.046 | 20899 |
| RLD-40 | 72.31 | 0.9998 | 0.035 | 20899 |
| **XLFM-Former (Ours)** | **52.34** | **0.9955** | **48.44** | **2631** |

Despite the numerical advantage of RLD at high iteration counts, our method achieves more than $700\times$ faster inference and $8\times$ lower memory consumption, while maintaining high structural fidelity. This highlights XLFM-Former's practicality for large-scale or real-time microscopy reconstruction.

## F.4 Inference Efficiency and Practical Deployment

To evaluate the practical efficiency of our method, we benchmarked XLFM-Former against four widely used backbone models on a single NVIDIA A100 GPU (80 GB), reporting inference speed (FPS), peak memory usage, and reconstruction accuracy (PSNR). As shown in Table 11, XLFM-Former achieves 48.44 FPS, well above the 30 FPS real-time threshold, while maintaining competitive memory consumption (2631 MiB) and the highest reconstruction fidelity (PSNR 52.34 dB) among all compared methods.

These results demonstrate that XLFM-Former is not only accurate but also efficient enough for real-time and high-throughput 3D microscopy applications.

Table 11: Inference efficiency and reconstruction accuracy on the H2B-Nemos dataset (single A100 GPU).

| Method | FPS ↑ | Peak Memory ↓ (MiB) | PSNR ↑ |
|---|---|---|---|
| ConvNeXt | 98.56 | 2565 | 50.88 |
| ViT | 94.79 | 2875 | 50.87 |
| U-Net | 74.34 | 2423 | 51.00 |
| ResNet-50 | 41.34 | 6467 | 51.34 |
| **XLFM-Former (Ours)** | **48.44** | **2631** | **52.34** |

XLFM-Former provides an effective balance between reconstruction fidelity, runtime speed, and memory efficiency, making it suitable for real-time 3D reconstruction in large-scale biological imaging systems. Qualitative comparisons in Figure 8 further demonstrate that applying the ORC

Loss suppresses background hallucinations and yields clearer neuronal boundaries across multiple orthogonal planes (XY, XZ, YZ).

### F.5  Extended analysis of ORC Loss

To further validate the contribution of the Optical Reconstruction Consistency (ORC) Loss, we provide a detailed quantitative comparison across multiple held-out zebrafish samples from the H2B-G8S and H2B-Nemos datasets. The ORC Loss encourages consistency between the forward projections of reconstructed and ground-truth volumes through the PSF, effectively constraining hallucinated structures. As shown in Table 12, incorporating the ORC (PSF) Loss consistently improves both PSNR and SSIM across all samples, confirming that it reduces reconstruction artifacts and enhances cross-sample generalization, particularly in challenging anatomical regions such as the hindbrain.

Table 12: Quantitative evaluation of ORC (PSF) Loss on independent datasets. Incorporating the loss improves both fidelity (PSNR) and perceptual quality (SSIM).

| Sample | PSNR$_{w/ PSF}$ | PSNR$_{w/o PSF}$ | SSIM$_{w/ PSF}$ | SSIM$_{w/o PSF}$ | $\Delta$PSNR | $\Delta$SSIM |
|---|---|---|---|---|---|---|
| G8S-#1 | 52.97 | 51.83 | 0.9913 | 0.9899 | +1.14 | +0.0014 |
| G8S-#2 | 59.34 | 59.13 | 0.9957 | 0.9955 | +0.21 | +0.0002 |
| G8S-#3 | 48.27 | 47.00 | 0.9894 | 0.9879 | +1.27 | +0.0015 |
| G8S-#4 | 54.12 | 53.90 | 0.9944 | 0.9940 | +0.21 | +0.0005 |
| G8S-#5 | 54.17 | 53.73 | 0.9949 | 0.9942 | +0.43 | +0.0008 |
| G8S-#6 | 48.88 | 47.26 | 0.9931 | 0.9930 | +1.62 | +0.0002 |
| Nemos-#1 | 52.88 | 52.17 | 0.9942 | 0.9914 | +0.71 | +0.0028 |
| Nemos-#2 | 50.62 | 49.55 | 0.9928 | 0.9892 | +1.07 | +0.0030 |

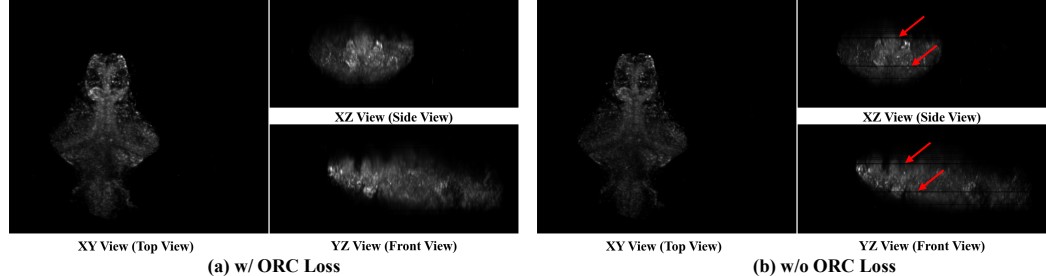

Figure 8: **Qualitative comparison of reconstructions with and without the ORC Loss.** Each column shows orthogonal views (XY, XZ, YZ) of the same zebrafish sample. The ORC Loss visibly suppresses background hallucinations and enhances structural clarity, especially in the hindbrain.

## G  Future Work

In addition, several preliminary experiments indicate that satisfactory reconstruction quality can still be achieved even when only a subset of microlenses (i.e., partial angular views) is used. This observation opens up another promising research direction: (1) how to adaptively determine the optimal number of lenses or sub-aperture views to balance throughput and reconstruction fidelity; and (2) whether a subset of the remaining views could be reserved as an evaluation signal to serve as an absolutely accurate ground truth for self-supervised or cross-view validation.

Beyond these specific directions, our findings suggest that future XLFM reconstruction research may benefit from a dual emphasis on *physical interpretability* and *functional usability* bridging optical modeling with neuroscience analysis in a unified framework.

