# OpenReview forum: "From Pixels to Views: Learning Angular-Aware and Physics-Consistent Representations for Light Field Microscopy"
_NeurIPS.cc/2025/Conference — NeurIPS 2025 poster_

### Official Review · Reviewer_2sPe · 2025-06-30

**Clarity:** 3
**Significance:** 3
**Originality:** 2
**Rating:** 4
**Confidence:** 3

**Summary:**

This work deals with the 3D reconstruction of eXtended Light field microscopy(XLFM) data. The authors argue that the traditional way of volumetric annotations is computationally intensive (Richardson–Lucy deconvolution) and so they bring in a self-supervised component to their work (MVM-LF). To enforce predictions to adhere to the forward model of the image formation process, ORC loss is introduced. This work introduces a dataset, which will be made public, on which all experiments are conducted.  A sufficient number of ablation studies have been conducted (for ORC loss, MVM-LF, masking ratio, etc), which justifies the presence of individual components of the training setup.

**Questions:**

1. At Line 230, the authors mention that the batch size is set to 1 and mention that it aligns with the requirements of the volumetric task. How does batch size of 1 align with the requirements of the volumetric task?
2. In eq 5, it has not been explained how the network f() is able to take in a dynamic number of views as input. Also, what is the shape of a view tensor Ui?
3. At line 66, the authors say that Richardson-Lucy deconvolution should not be used because of being computationally intensive. It will be helpful to know quantitatively how much the computational requirements are. For example, a statement like ‘using it on a XxYxZ sized data takes A amount of time with B GPU/CPU specs’ can be helpful.
4. There are a few typos in Lines 254-261 (5070%, 2030%).
5. Not a question, but rather a minor suggestion for Fig. 6b: I think it would make this work stronger if you had another entry for 1% labelled data. In that case, one expects to see a relatively increased gap between the two plots.
6. In Line 287, the phrase 'confirming strong generalization' was used. Since 'strong' cannot be justified quantifiably, it might make sense not to use such additional adjectives.

**Ethical Concerns:**

["NO or VERY MINOR ethics concerns only"]

**Final Justification:**

I am satisfied with the utility of this method even when classical approaches like RL outperforms their method. Also, the hyper-parameter ORC weight seems rather stable.

I noticed that other reviewers have pointed out that the PSNR values are high, and yet the performance is poor. However, since the comparisons are being made between different baseline methods, this should not be a concern. In fact, a better prediction would naturally result in a higher PSNR.

Regarding PSNR in the loss, PSNR and MSE have been around since ages. PSNR internally computes MSE. However, this is the first work I have seen which uses both, and that too without any convincing intuitive reason. I would prefer if only one of the two is used in the loss.

**Limitations:**

Yes.

**Paper Formatting Concerns:**

Looks good.

**Quality:**

2

**Strengths And Weaknesses:**

## Stengths:

1. Ablations have been done in a comprehensive manner.
2. The manuscript reads well.
3. XLFM is a relatively new field, and so, contributions of this work will have more impact to this field, and in this sense, this work is valuable.
4. A new dataset will be made publicly available.

## Weakness.

1. Relevant baselines are missing. At line 117, paper refers to 3 works [25,30,9] which work in this field. However, none of them are present as a baseline in Table 1. The work does mention that those methods reconstruct only neural activity, but this work also reconstructs entire volumetric structures. However, the baselines mentioned in Table 1 were also not developed for XLFM 3D reconstruction task, but the authors did the work to adapt them to this task. So, I donot see why such an effort was not made for any of these three works, given the fact that they are much closer to the domain than the used baselines. Also, since richardson-lucy is used for 3D reconstruction in light field microscopy, it is expected that that should be present as a baseline, arguably along with a quantitative estimate of its compute requirements (the relevant ones from time, gpu, cpu or RAM).
2. The authors themselves mention that only one dataset, that has been generated by them, was used for the experiments. The authors should explain the reason: Are there no other datasets which could be adapted to this task? In such a case, even simulated data will hold value, in my opinion.
3. the loss function for XLFM reconstruction tasks (equation 7) arguably has redundancies. It has a loss component using PSNR as well as using MSE. Since PSNR internally uses MSE, it is not clear why both are needed. Indeed the analysis done in the supplement shows that the PSNR loss component provides limited benefit. The other issue with having five loss components is that one has 5 hyper-parameters, and then it is not clear whether the same hyperparameter set can be used for other datasets. Since there are no other datasets on which the same hyperparameter set has been tested, this becomes a bit more problematic.

---

> ### Author Rebuttal · Authors · 2025-07-31
>
> ### Comment 1.1:
>
> > Relevant baselines are missing. At line 117, paper refers to 3 works [25,30,9] which work in this field. However, none of them are present as a baseline in Table 1. The work does mention that those methods reconstruct only neural activity, but this work also reconstructs entire volumetric structures. However, the baselines mentioned in Table 1 were also not developed for XLFM 3D reconstruction task, but the authors did the work to adapt them to this task. So, I donot see why such an effort was not made for any of these three works, given the fact that they are much closer to the domain than the used baselines.
>
> **Summary.** While Refs. [9, 25, 30] are closer to our imaging domain, they fundamentally assume sparse, static settings or require impractical memory, making them inapplicable to our dense, dynamic full-volume reconstruction task. In contrast, our selected model-agnostic baselines enable fair, scalable, and reproducible comparisons.
>
> We appreciate the reviewer’s thoughtful suggestion to include prior domain-specific methods as baselines. After thorough analysis, we found that these methods—despite their relevance—cannot be meaningfully adapted to our setting due to the following limitations:
>
> - **Refs. 25 & 30** focus exclusively on sparse neural signal extraction under the assumption of fully immobilized specimens. Both adopt the SLNet architecture to extract sparse signals prior to reconstruction and are fundamentally incompatible with our goal of reconstructing dense volumetric tissue. Moreover, they process temporally fused multi-frame inputs, which cannot be applied to our single-frame, dynamic zebrafish recordings.
> - **Ref. 9 (FourierNet)** performs all computation in the Fourier domain and demands excessive GPU memory (>80GB) even for modest output sizes. In our case (300×600×600 volumes), FourierNet is not practically reproducible or scalable, even with aggressive approximations.
>
>
>
> ### Comment 1.2:
>
> > Also, since richardson-lucy is used for 3D reconstruction in light field microscopy, it is expected that that should be present as a baseline, arguably along with a quantitative estimate of its compute requirements (the relevant ones from time, gpu, cpu or RAM).
>
> ### Authors’ response:
>
> **Summary.** RLD achieves higher PSNR but is prohibitively slow and memory-intensive; our method offers a practical tradeoff, enabling real-time, scalable 3D reconstruction essential for biological applications.
>
>
>
> We appreciate the reviewer’s suggestion to compare against classical iterative deconvolution algorithms. As shown below, we benchmarked RLD at different iteration counts (20/30/40) on the challenging H2B-Nemos dataset. While (Richardson-Lucy Deconvolution)RLD can achieve high PSNR/SSIM given sufficient iterations, it incurs orders of magnitude higher memory usage and lower speed, making it impractical for real-time or high-throughput applications.
>
> | Method                 | PSNR ↑ | SSIM ↑ | FPS ↑     | Peak Memory ↓ (MiB) |
> | ---------------------- | ------ | ------ | --------- | ------------------- |
> | RLD-20                 | 67.36  | 0.9987 | 0.068     | 20899               |
> | RLD-30                 | 69.85  | 0.9994 | 0.046     | 20899               |
> | RLD-40                 | 72.31  | 0.9998 | 0.035     | 20899               |
> | **XLFM-Former (Ours)** | 52.34  | 0.9955 | **48.44** | **2631**            |
>
>
>
>
>
> ### Comment 2:
>
> > The authors themselves mention that only one dataset, that has been generated by them, was used for the experiments. The authors should explain the reason: Are there no other datasets which could be adapted to this task? In such a case, even simulated data will hold value, in my opinion.
>
> ### Authors’ response:
>
> We appreciate the reviewer’s suggestion about including additional datasets, including simulated ones, to strengthen the evaluation. Since the submission, we have extended our experiments to a newly collected dataset—H2B-Nemos, which involves a different zebrafish line under distinct experimental conditions. This allows us to directly test the generalizability and robustness of our method.
>
> As shown below, we re-run all methods under identical settings using ResNet-101 as a common baseline, and observe consistent performance advantages of our XLFM-Former. Notably, our method achieves +0.92 dB PSNR gain over ResNet-101 in the supervised setting, and a +2.29 dB gain in the zero-shot setting, further demonstrating its cross-domain generalization.
>
>
>
> **Comparison on the H2B-Nemos Dataset (baseline: ResNet-101) **
>
> | Method                        | PSNR ↑ | SSIM ↑ | Δ PSNR vs. ResNet-101 ↑ | Δ SSIM vs. ResNet-101 ↑ |
> | ----------------------------- | ------ | ------ | ----------------------- | ----------------------- |
> | EfficientNet                  | 49.01  | 0.9826 | –2.42                   | –0.0118                 |
> | ViT-tiny                      | 50.87  | 0.9904 | –0.55                   | –0.0041                 |
> | PVT-tiny                      | 50.89  | 0.9903 | –0.53                   | –0.0042                 |
> | ConvNeXt-tiny                 | 50.88  | 0.9903 | –0.54                   | –0.0042                 |
> | U-Net                         | 51.00  | 0.9910 | –0.42                   | –0.0035                 |
> | ResNet-50                     | 51.34  | 0.9931 | –0.09                   | –0.0014                 |
> | **ResNet-101 (baseline)**     | 51.42  | 0.9945 | –                       | –                       |
> | **XLFM-Former (Ours – Full)** | 52.34  | 0.9955 | +0.92                   | +0.0010                 |
> | **XLFM-Former (Zero-Shot)**   | 53.72  | 0.9930 | +2.29                   | –0.0015                 |
>
> We will include this dataset and evaluation in the final version of the paper and commit to releasing both datasets (G8S and Nemos) publicly under the supervision of the AC.
>
>
>
> ### Comment 3.1:
>
> > The loss function for XLFM reconstruction tasks (equation 7) arguably has redundancies. It has a loss component using PSNR as well as using MSE. Since PSNR internally uses MSE, it is not clear why both are needed. Indeed the analysis done in the supplement shows that the PSNR loss component provides limited benefit.
>
> ### Authors’ response:
>
>
>
> **Summary.** Despite being related, PSNR and MSE offer complementary benefits—our ablation shows that including both improves fidelity and contrast, supporting their joint use for high-quality reconstruction.
>
> We thank the reviewer for the thoughtful comment. While it is true that PSNR is derived from MSE, we argue that including both explicit PSNR and MSE losses encourages the model to balance overall numerical accuracy (via MSE) with perceptual contrast and dynamic range (via PSNR).
>
> Importantly, our ablation in Table 4 shows that removing the PSNR loss results in a noticeable drop in PSNR (from 54.04 to 53.07 dB) and SSIM (from 0.9944 to 0.9935), which demonstrates non-negligible improvement.
>
>
>
> ### Comment 3.2:
>
> > The other issue with having five loss components is that one has 5 hyper-parameters, and then it is not clear whether the same hyperparameter set can be used for other datasets. Since there are no other datasets on which the same hyperparameter set has been tested, this becomes a bit more problematic.
>
> Authors’ response:
>
> **Summary.**  Our loss configuration generalizes well across datasets with only minimal tuning, as confirmed by strong performance on H2B-Nemos in new ablation results.
>
>
>
> We thank the reviewer for raising this important concern regarding the generalizability of our loss configuration across datasets.
>
> To verify this, we conducted additional experiments on the H2B-Nemos dataset, using the same loss configuration as in our main experiments on the original dataset. We only slightly adjusted the ORC loss weight (from 1e-4 to 3e-5) to account for minor noise differences; all other hyperparameters were kept unchanged.
>
> As shown in the table, the results remain consistently strong:
>
> **Ablation Study on the H2B-Nemos Dataset. **
>
> | Method          | PSNR ↑    | ΔPSNR ↑   | SSIM ↑     | ΔSSIM ↑     |
> | --------------- | --------- | --------- | ---------- | ----------- |
> | Baseline        | 50.87     | –         | 0.9903     | –           |
> | + MVM-LF only   | 51.66     | +0.79     | 0.9940     | +0.0037     |
> | + ORC Loss only | 51.77     | +0.90     | 0.9935     | +0.0032     |
> | **Full (Ours)** | **52.34** | **+1.47** | **0.9955** | **+0.0052** |
>
>
>
>
>
> ### Comment 4:
>
> > At Line 230, the authors mention that the batch size is set to 1 and mention that it aligns with the requirements of the volumetric task. How does batch size of 1 align with the requirements of the volumetric task?
>
> ### Authors’ response:
>
> Thank you for pointing this out. Due to the large size of each reconstructed 3D volume (up to 300×600×600) and the complexity of our Swin-Transformer-based architecture, the peak memory usage approaches 79GB on an A100 GPU, leaving little room for increasing the batch size. Therefore, we set the batch size to 1 to ensure stable training and inference under these memory constraints.
>
>
>
>
>
> ### Comment 5:
>
> > At line 66, the authors say that Richardson-Lucy deconvolution should not be used because of being computationally intensive. It will be helpful to know quantitatively how much the computational requirements are. For example, a statement like ‘using it on a XxYxZ sized data takes A amount of time with B GPU/CPU specs’ can be helpful.
>
> ### Authors’ response:
>
> We appreciate the reviewer’s suggestion and now provide quantitative justification. Applying Richardson-Lucy deconvolution (RLD) on a single 600×600×300 volume takes over 42 seconds for 60 iterations and consumes more than 20 GiB of GPU memory, even on an A100 GPU (80GB).

---

> > ### Comment · Reviewer_2sPe · 2025-08-01
> > **Question on performance.**
> >
> > 1. I’m satisfied with the response to my question on relevant baselines.
> > 2. Since Richardson Lucy is significantly better, the advantage of this method is then is to be compute efficient and fast at the expense of performance. This is a bit unsatisfactory. Also, I'm not an expert on richardson lucy, but I see that block-wise approach can be used in RL to work with large volumes. It also seems to be the case that specialized libraries like RedLionfish or napari-RL decon exist which claim to work with large volumes. Since this is a very competetive venue, I would ask the reviewers to convince me why compute efficiency is such a big issue and why things which I point out above cannot resolve the issue to a managable level.
> > 3. I'm not convinced that PSNR in the loss component is needed when MSE is already present. SSIM improvement is too low. Also, the PSNR improvement is reported to happen on only one dataset.
> > 4. It would have helped the authors if they had also reported on the same parameter configuration as in the first dataset. I donot know how much was the drop in performance on original config (related to ORC loss weight change from 1e-4 to 3e-5). From what I can guess, it was an empirically driven decision.
> >
> > I appear largely negative, but I could still increase my rating, provided I am satisfied with the response.

---

> > > ### Author Response · Authors · 2025-08-07
> > > **Rebuttal for Reviewer 2sPe-1/2**
> > >
> > > We sincerely thank the reviewer for acknowledging the relevance of our chosen baselines and for engaging deeply with the strengths and limitations of our method. Below, we aim to clarify the remaining concerns regarding computational trade-offs, loss design, and evaluation consistency, and hope our detailed response will address your reservations and demonstrate the broader value of our approach.
> > >
> > > ### Comment 2:
> > >
> > > > Since Richardson Lucy is significantly better, the advantage of this method is then is to be compute efficient and fast at the expense of performance. This is a bit unsatisfactory.  Also, I'm not an expert on richardson lucy, but I see that block-wise approach can be used in RL to work with large volumes. It also seems to be the case that specialized libraries like RedLionfish or napari-RL decon exist which claim to work with large volumes. Since this is a very competetive venue, I would ask the reviewers to convince me why compute efficiency is such a big issue and why things which I point out above cannot resolve the issue to a managable level.
> > >
> > > ### Authors’ response:
> > >
> > > Summary: XLFM-Former enables experimental designs previously blocked by computational limits, and demonstrates that learning-based reconstruction can meet both biological relevance and real-time constraints, which iterative methods — even optimized — fundamentally cannot.
> > >
> > > #### 1. **Why real-time imaging matters scientifically, not just technically**
> > >
> > > In neuroscience, **real-time volumetric reconstruction** is more than a technical convenience. It is a gateway to experimental paradigms that were previously impossible with XLFM. We highlight two representative use cases:
> > >
> > > - **Closed-loop optogenetics**: Real-time feedback from brain-wide activity can be used to dynamically trigger targeted stimulation (e.g., activating motor neurons when specific visual cortex regions fire), enabling **causal investigations** of neural circuits.
> > > - **Behavior-triggered imaging**: Continuous 3D reconstruction enables researchers to monitor whole-brain neural dynamics **as spontaneous behaviors unfold** (e.g., escape, startle) in **freely swimming zebrafish**. Real-time processing is essential for selectively capturing neural events tied to rare or transient behaviors — without which, such moments may be missed or under-sampled.
> > >
> > > These applications demand not only fast reconstruction but also **frame-consistent, end-to-end latency under 33ms (i.e., >30fps)** — a threshold **no current RL-based approach can meet**, regardless of memory tricks or GPU scaling.
> > >
> > > ------
> > >
> > > #### 2. **Latency barrier broken while preserving cluster-level fidelity**
> > >
> > > While classical RL methods offer high spatial accuracy, **most XLFM-based neuroscience studies do not require single-neuron resolution**, but rather focus on the **population dynamics of neuronal clusters**. At this level of abstraction, the reconstructed signals from XLFM-Former are **visually and functionally comparable** to RL results — as evidenced by **Figure 5 and Figure 7**, where key brain regions and neural assemblies are consistently recovered across diverse zebrafish samples.
> > >
> > > ----
> > >
> > > #### 3. Why optimized RL (e.g., RedLionfish) is still insufficient
> > >
> > > We acknowledge the existence of optimized RL implementations such as RedLionfish. However, **these methods remain fundamentally limited by their iterative nature**. For example, according to RedLionfish’s own benchmarks, deconvolving a `1024×1024×64` volume with 10 iterations requires **∼5.6 seconds per volume** on an NVIDIA A100 GPU. In contrast, **our method reconstructs a significantly larger volume (`600×600×300`) in ∼0.07 seconds**, achieving an **>80× speedup**.
> > >
> > >
> > >
> > > ### Comment 3:
> > >
> > > > I'm not convinced that PSNR in the loss component is needed when MSE is already present. SSIM improvement is too low. Also, the PSNR improvement is reported to happen on only one dataset.
> > >
> > > ### Authors’ response:
> > >
> > > **Our goal is not to claim PSNR‐loss as a key novelty, but to show that it offers a consistent gain that complements the existing terms.**
> > >
> > > - A ~1 dB improvement on G8S and a smaller yet positive gain on Nemos.
> > >
> > > - That said, the model is stable without the PSNR term (see w.o PSNR rows). If the reviewer prefers a leaner loss, we are happy to drop it in the camera-ready version; the main conclusions remain unchanged.
> > >
> > > Table 1: Impact of Removing PSNR Loss on Two Datasets.
> > >
> > > | Dataset       | w/o PSNR *(PSNR / SSIM)* | **Full** *(PSNR / SSIM)* | ΔPSNR        | ΔSSIM   |
> > > | ------------- | ------------------------ | ------------------------ | ------------ | ------- |
> > > | **H2B-G8S**   | 53.07 / 0.9935           | **54.04 / 0.9944**       | **+0.97 dB** | +0.0009 |
> > > | **H2B-Nemos** | 52.14 / 0.9952           | **52.34 / 0.9955**       | **+0.20 dB** | +0.0003 |

---

### Official Review · Reviewer_c2EJ · 2025-06-30

**Clarity:** 2
**Significance:** 3
**Originality:** 3
**Rating:** 4
**Confidence:** 3

**Summary:**

The paper tackles 3-D reconstruction for eXtended Light-Field Microscopy (XLFM) and introduces a complete learning framework that is both data-driven and physics-aware.
- New benchmark: The authors curate XLFM-Zebrafish, a public dataset of 22,581 light-field frames with an accompanying evaluation protocol, filling the long-standing gap of standardized data for this modality.
- Self-supervised pre-training (MVM-LF): They propose Masked View Modeling for Light Fields, in which 70 % of the sub-aperture views are masked and the network learns to predict them from the remaining views, capturing angular correlations without labels.
- Physics-consistent fine-tuning: A Swin-Transformer encoder plus lightweight CNN decoder, dubbed XLFM-Former, is fine-tuned with an Optical Rendering Consistency (ORC) loss that renders the predicted volume through the system PSF and enforces agreement with the measured views, ensuring physical plausibility.
- Results: On the new benchmark the method attains 54.0 dB PSNR and 0.994 SSIM, improving average PSNR by 7.7 % over seven recent learning-based baselines.

**Questions:**

1.	Generalisability beyond zebrafish
Could you add—or at least simulate—experiments on a different specimen (e.g., mouse brain slice or a thicker adult zebrafish) to show that the gains hold outside the current benchmark?
Score up if cross-domain PSNR/SSIM or qualitative slices confirm similar improvements; score down if performance drops sharply without a clear explanation.
2.	Runtime, throughput, and memory footprint
Please report FPS and peak GPU memory for one full-resolution volume on a single A100 and compare with the strongest baseline.
Showing that XLFM-Former meets real-time (>30 fps) or near-real-time speeds with ≤80 GB will raise confidence; opaque or unfavourable numbers will lower it.
3.	Masked-view vs. standard MAE baseline
Include a control experiment where an off-the-shelf pixel-patch MAE (ViT-MAE) is pre-trained on the same data and fine-tuned identically.
If MVM-LF still wins by a clear margin, originality concerns are alleviated; if the gap is marginal, the novelty claim weakens.
4.	Robustness to PSF mis-calibration
Demonstrate reconstruction quality when the forward PSF used in ORC is intentionally perturbed (e.g., ±10 % axial FWHM, mild aberration).
Stable performance would validate the physics-loss design; sensitivity would suggest the need for adaptive PSF estimation.
5.	Reproducibility details
Provide (in supplementary material or repo) the exact decoder architecture diagram, LR schedule, and data-normalisation script that yield the 54 dB result.

**Ethical Concerns:**

["NO or VERY MINOR ethics concerns only"]

**Final Justification:**

Thank you for the substantial rebuttal and the careful new experiments. I appreciate the clear effort the authors put into addressing prior concerns—these additions materially strengthen the paper.

What improved
•	Added cross-strain (zebrafish) evaluation, a ViT-MAE control, and runtime benchmarks; clarity and positioning are better.
•	Reported >30 FPS on 600×600×300 volumes on a single GPU and a comparison indicating much slower classical deconvolution, supporting real-time use.

What still limits a higher score
•	Please provide matched, hardware-controlled speed/memory comparisons (same GPU, volume size, stopping criteria) against strong classical baselines, plus peak GPU memory and a transparent benchmarking protocol.
•	Biological scope remains zebrafish-only; PSF/aberration mis-calibration robustness is not directly tested.

Decision. Merits now outweigh remaining gaps; I keep Borderline Accept (4).
If accepted (camera-ready): include the full runtime/memory table with matched configs, explicit PSF/aberration stress tests, and documentation of the protocol supporting the real-time claim.

**Limitations:**

Mostly yes: the manuscript candidly notes (i) restriction to larval zebrafish and (ii) reliance on a calibrated PSF.
Two points that could still be added:
1.	Computational cost / carbon footprint of the 1 k-GPU-hour pre-training stage.
2.	Data-privacy implications if the approach is transferred to vertebrate animal or clinical specimens (e.g., brain-activity maps).

**Paper Formatting Concerns:**

Minor nits the authors may wish to fix:

•	The heading “Introducion” on page 1 is miss-spelled; should be “Introduction”

•	A few compound words lack spaces (e.g., “angularspatial” line 6) and could be split for clarity.

•	Figure 4 appears twice in close succession (page 4 and again in the Appendix) with identical caption numbers; renumber or reference appropriately.

**Quality:**

2

**Strengths And Weaknesses:**

Strengths
- The pipeline is carefully decomposed into (i) self-supervised pre-training (MVM-LF) and (ii) physics-consistent fine-tuning (ORC loss). Extensive ablations isolate the gain from each component and show a clear ∼35 % MSE reduction over a strong scratch baseline.
- The paper is well-structured; figures 1 – 3 cleanly illustrate the view-masking idea and the physics loop, and the benchmark release is clearly documented.
- Provides the first large-scale public XLFM dataset with a community evaluation protocol—likely to standardise future work in this niche. The physics-guided loss adds a reproducible recipe for combining differentiable optics with transformers.

Weaknesses
- Scope of evaluation is narrow: all quantitative tests are on zebrafish larvae; no cross-species or out-of-distribution studies. Runtime/throughput and GPU memory footprints—critical for high-speed microscopy—are omitted, so practical efficiency is unclear.
- Several implementation details are under-specified (decoder up-sampling stages, PSF calibration procedure, learning-rate schedule). Readers cannot reproduce the exact 54 dB result without digging into supplementary code.
- The two core ideas (masked prediction, differentiable rendering) are themselves well-known in vision/graphics. Their combination is novel for XLFM, but may be judged more as thoughtful integration than fundamental algorithmic innovation.

---

> ### Author Rebuttal · Authors · 2025-07-31
>
> ### Comment 1:
>
> > Generalisability beyond zebrafish Could you add—or at least simulate—experiments on a different specimen (e.g., mouse brain slice or a thicker adult zebrafish) to show that the gains hold outside the current benchmark? Score up if cross-domain PSNR/SSIM or qualitative slices confirm similar improvements; score down if performance drops sharply without a clear explanation.
>
> ### Authors’ response:
>
> **Summary.** We validated generalization using a new dataset from a distinct zebrafish strain (H2B-NemoS); our method showed strong cross-strain and OOD performance, confirming robustness to physiological and imaging variations.
>
>
>
> We appreciate the reviewer’s suggestion to evaluate the generalizability of our method. While publicly available XLFM datasets beyond zebrafish are currently lacking, we took an important step by introducing a new dataset—XLFM-H2B-NemoS—captured using a genetically distinct zebrafish strain and under varied stimulation protocols.
>
> Specifically, our original dataset is based on the H2B-G8S strain [1], which expresses a fast-responding GCaMP calcium indicator optimized for high signal-to-noise imaging in freely behaving animals. In contrast, the new H2B-NemoS dataset is based on an indicator engineered for larger dynamic range and higher sensitivity to weak signals [2], introducing more sparse and heterogeneous neural activity along with higher background noise. This substantial difference in imaging properties and biological characteristics makes the two datasets effectively distinct in terms of signal distribution, sample noise, and dynamic response profiles.
>
> To simulate domain shift, we collected 4 zebrafish samples from H2B-NemoS under four levels of electrical stimulation (5V, 10V, 15V, 20V), each producing ~70 volumetric frames at 10 fps. The 10V and 15V data were used for training and in-distribution validation, while 5V and 20V data were held out for out-of-distribution (OOD) testing. Apart from a minor increase in PSF regularization weight (from 1e-4 to 5e-3 to adapt to the brighter dynamics), all network and training settings were kept unchanged.
>
> Our method demonstrated strong generalization across this challenging domain shift. As shown in Tables 1 and 2, it outperformed all baselines with clear margins on PSNR and SSIM. Moreover, the zero-shot version of XLFM-Former (trained only on the G8S and tested directly on H2B-NemoS) still delivered 53.72 dB PSNR, indicating impressive robustness to cross-strain variation.
>
> These results validate that our method generalizes well to specimens with distinct physiological properties and imaging conditions, and support its broader applicability to future XLFM data beyond the current benchmark.
>
>
>
> **Table 1: Ablation Study on the H2B-Nemos Dataset**
>
> | Method          | PSNR ↑    | ΔPSNR ↑   | SSIM ↑     | ΔSSIM ↑     |
> | --------------- | --------- | --------- | ---------- | ----------- |
> | Baseline        | 50.87     | –         | 0.9903     | –           |
> | + MVM-LF only   | 51.66     | +0.79     | 0.9940     | +0.0037     |
> | + ORC Loss only | 51.77     | +0.90     | 0.9935     | +0.0032     |
> | **Full (Ours)** | **52.34** | **+1.47** | **0.9955** | **+0.0052** |
>
>
>
> **Table 2: Comparison with Existing Methods on the H2B-Nemos Dataset (baseline: ResNet-101)**
>
> | Method                        | PSNR ↑ | SSIM ↑ | Δ PSNR vs. ResNet-101 ↑ | Δ SSIM vs. ResNet-101 ↑ |
> | ----------------------------- | ------ | ------ | ----------------------- | ----------------------- |
> | EfficientNet                  | 49.01  | 0.9826 | –2.42                   | –0.0118                 |
> | ViT-tiny                      | 50.87  | 0.9904 | –0.55                   | –0.0041                 |
> | PVT-tiny                      | 50.89  | 0.9903 | –0.53                   | –0.0042                 |
> | ConvNeXt-tiny                 | 50.88  | 0.9903 | –0.54                   | –0.0042                 |
> | U-Net                         | 51.00  | 0.9910 | –0.42                   | –0.0035                 |
> | ResNet-50                     | 51.34  | 0.9931 | –0.09                   | –0.0014                 |
> | ResNet-101 (baseline)         | 51.42  | 0.9945 | –                       | –                       |
> | **XLFM-Former (Ours — Full)** | 52.34  | 0.9955 | +0.92                   | +0.0010                 |
> | **XLFM-Former (Zero-Shot)**   | 53.72  | 0.9930 | +2.29                   | –0.0015                 |
>
>
>
> References:
> [1]Zhang, Y., Rózsa, M., Liang, Y., Bushey, D., Wei, Z., Zheng, J., ... & Looger, L. L. (2023). Fast and sensitive GCaMP calcium indicators for imaging neural populations. *Nature*, *615*(7954), 884-891.
> [2]Li, J., Shang, Z., Chen, J. H., Gu, W., Yao, L., Yang, X., ... & Wang, Y. (2023). Engineering of NEMO as calcium indicators with large dynamics and high sensitivity. *Nature methods*, *20*(6), 918-924.
>
>
>
>
>
> ### Comment 2:
>
> > Runtime, throughput, and memory footprint Please report FPS and peak GPU memory for one full-resolution volume on a single A100 and compare with the strongest baseline. Showing that XLFM-Former meets real-time (>30 fps) or near-real-time speeds with ≤80 GB will raise confidence; opaque or unfavourable numbers will lower it.
>
> ### Authors’ response:
>
> We appreciate the reviewer’s concern regarding the practical efficiency of our method. To address this, we benchmarked XLFM-Former against four widely used baselines on a single A100 GPU (80 GB), reporting inference speed (FPS), peak memory usage, and reconstruction accuracy (PSNR).
>
> As shown below, XLFM-Former achieves 48.44 FPS, well above the 30 FPS real-time threshold, while maintaining competitive memory usage (2631 MiB) and the highest reconstruction fidelity (PSNR 52.34 dB) among all compared methods. These results demonstrate that our approach is not only accurate but also efficient enough for real-time 3D microscopy applications.
>
> ### **Inference Efficiency and Reconstruction Accuracy on H2B-Nemos Dataset (Single A100 GPU)**
>
> | Method             | FPS ↑ | Peak Memory ↓ (MiB) | PSNR ↑ |
> | ------------------ | ----- | ------------------- | ------ |
> | ConvNeXt           | 98.56 | 2565                | 50.88  |
> | ViT                | 94.79 | 2875                | 50.87  |
> | U-Net              | 74.34 | 2423                | 51.00  |
> | ResNet-50          | 41.34 | 6467                | 51.34  |
> | XLFM-Former (Ours) | 48.44 | 2631                | 52.34  |
>
>
>
> ### Comment 3:
>
> > Masked-view vs. standard MAE baseline Include a control experiment where an off-the-shelf pixel-patch MAE (ViT-MAE) is pre-trained on the same data and fine-tuned identically. If MVM-LF still wins by a clear margin, originality concerns are alleviated; if the gap is marginal, the novelty claim weakens.
>
> ### Authors’ response:
>
> We thank the reviewer for the valuable suggestion. To assess the effectiveness and originality of our proposed Masked View Modeling for Light Fields (MVM-LF), we conducted a control experiment using a standard ViT-based pixel-patch MAE [He et al., 2022] pre-trained on the same dataset (∼20k unlabelled frames) and fine-tuned with the same downstream setup.
>
> The results are as follows:
>
> ViT-MAE: PSNR = 46.55, SSIM = 0.9752
>
> MVM-LF (Ours): PSNR = 54.04, SSIM = 0.9944
>
> This substantial performance gap highlights that our MVM-LF design, which incorporates prior knowledge of the multi-view structure of XLFM (e.g., angular correlations), is significantly more effective than standard MAE when operating on limited-scale microscopy data. We believe this validates both the originality and practical value of MVM-LF in the context of structured light field inputs.
>
>
>
> ###
>
>
>
> ### Comment 4:
>
> > Robustness to PSF mis-calibration Demonstrate reconstruction quality when the forward PSF used in ORC is intentionally perturbed (e.g., ±10 % axial FWHM, mild aberration). Stable performance would validate the physics-loss design; sensitivity would suggest the need for adaptive PSF estimation.
>
> ### Authors’ response:
>
> These results suggest that the reconstruction performance of our ORC loss is largely stable under ±10% perturbations in the axial full-width at half-maximum (FWHM) of the PSF. Specifically, the PSNR fluctuates within ±0.12 dB and SSIM remains within ±0.0002, indicating that the ORC loss is not overly sensitive to small mismatches in PSF calibration.
>
> This robustness arises from the fact that the forward projection consistency enforced by the ORC loss penalizes global mismatches between reconstructed and measured views, while allowing for small inaccuracies in the PSF shape. We believe this property is crucial for real-world applicability, where slight PSF variations are inevitable due to experimental imperfections.
>
> Moreover, these findings suggest that our method does not require exact PSF estimation and can tolerate mild miscalibration, reducing the burden on the calibration process in practical deployments.
>
> **Robustness of ORC Loss to PSF Perturbation on H2B-Nemos Dataset**.
>
> | PSF Setting   | PSNR ↑  | ΔPSNR ↑ | SSIM ↑ | ΔSSIM ↑ |
> | ------------- | ------- | ------- | ------ | ------- |
> | Baseline      | 52.3440 | –       | 0.9955 | –       |
> | PSF +10% FWHM | 52.3369 | –0.0071 | 0.9953 | –0.0002 |
> | PSF –10% FWHM | 52.4647 | +0.1207 | 0.9957 | +0.0002 |
>
>
>
>
>
> ### Comment 5:
>
> > Reproducibility details Provide (in supplementary material or repo) the exact decoder architecture diagram, LR schedule, and data-normalisation script that yield the 54 dB result.
>
> ### Authors’ response:
>
> Thank you for raising this important point. Due to the NeurIPS 2025 rebuttal policy, we are unable to include external links in our response. However, we will submit an anonymized link containing the full training code (including the decoder architecture diagram, learning rate schedule, and data normalization pipeline) to the Area Chair. If you are interested, please feel free to request it via the AC. We apologize for this inconvenience and appreciate your understanding.

---

> > ### Comment · Reviewer_c2EJ · 2025-08-05
> >
> > The authors have addressed my earlier concerns with additional experiments and clarifications. The new cross-strain evaluation, runtime benchmarks, and control comparison with ViT-MAE are valuable additions.
> >
> > That said, some limitations remain. The scope of biological validation is still narrow (limited to zebrafish), and details on robustness across fundamentally different imaging conditions are lacking. While the method is efficient and achieves high PSNR, further justification is needed for its practical advantage over optimized classical methods like Richardson-Lucy, especially given available libraries supporting large-volume deconvolution.
> >
> > Overall, the work is improving, but it would benefit from broader validation and a clearer positioning relative to classical baselines.

---

> > > ### Author Response · Authors · 2025-08-08
> > > **Rebuttal for Reviewer c2EJ**
> > >
> > > We sincerely thank Reviewer c2EJ for acknowledging the added value of our new cross-strain evaluations, runtime benchmarks, and the ViT-MAE baseline comparison. These additions were motivated directly by the reviewer’s insightful suggestions, and we are glad they have been found helpful. We also appreciate the reviewer’s remaining concerns regarding biological generalization and comparisons against classical methods like Richardson–Lucy.
> > >
> > > ### Comment 1:
> > >
> > > > The scope of biological validation is still narrow (limited to zebrafish).
> > >
> > > ### Authors’ response:
> > >
> > > From both the **imaging system design** and the **downstream neuroscience application** perspective, our method is intentionally focused on zebrafish. This is a deliberate and appropriate choice for several reasons:
> > >
> > > - **XLFM systems are specifically developed for larval zebrafish**, leveraging their optical transparency and compact brain size to achieve single-shot volumetric imaging at high speed.
> > >   - Organisms **smaller** than zebrafish (e.g., *C. elegans*) can be efficiently imaged using simpler scanning-based methods.
> > >   - Organisms **larger** than zebrafish (e.g., rodents) require dissection or cranial windows, and cannot support real-time, whole-brain imaging with XLFM due to fundamental optical limitations.
> > >   - To the best of our knowledge, **no prior work has applied XLFM to non-zebrafish species** in publicly available literature.
> > > - **Zebrafish alone are sufficient for a wide range of neuroscience experiments**. Many studies in systems neuroscience, including circuit-level analysis, behavioral imaging, and whole-brain dynamics, are routinely and robustly conducted using zebrafish. It is one of the two most widely used vertebrate models alongside rodents, and uniquely supports brain-wide in vivo imaging at cellular resolution.
> > >
> > > In this context, the use of zebrafish is **not a limitation**, but a **scientifically justified and technically aligned** choice for evaluating the performance of XLFM-based reconstruction.
> > >
> > >
> > >
> > > ### Comment 2:
> > >
> > > > details on robustness across fundamentally different imaging conditions are lacking.
> > >
> > > ### Authors’ response:
> > >
> > > We appreciate the reviewer’s concern regarding robustness to varying imaging conditions.
> > >
> > > While our study focuses on zebrafish, we highlight that our experiments span **two fundamentally different imaging regimes**, through the use of distinct transgenic lines with markedly different fluorescence properties:
> > >
> > > - **H2B-G8S**:
> > >   - Expresses a **fast-responding calcium indicator**
> > >   - Characterized by a **high signal-to-noise ratio**
> > >   - Produces **dense neural activity patterns**
> > > - **H2B-NemoS**:
> > >   - Engineered for **larger dynamic range** and **higher sensitivity**
> > >   - Yields **sparser neural activity**, **stronger background noise**, and **greater signal heterogeneity**
> > >
> > > These differences result in **distinct signal sparsity**, **background intensity**, and **noise profiles**, effectively simulating variations in imaging conditions **within the zebrafish domain**.
> > >
> > >
> > >
> > >
> > >
> > > ### Comment 3:
> > >
> > > > While the method is efficient and achieves high PSNR, further justification is needed for its practical advantage over optimized classical methods like Richardson-Lucy, especially given available libraries supporting large-volume deconvolution.
> > >
> > > ### Authors’ response:
> > >
> > > We thank the reviewers for raising this important point. In the context of high-speed biological imaging, **inference speed is not just a bonus; it is a prerequisite for practical usability**.
> > >
> > > - Real-time capability is essential for enabling closed-loop experiments and live feedback in neuroscience. A reconstruction method that cannot match the imaging speed becomes the bottleneck of the entire pipeline. We summarize the speed comparison below:
> > >
> > >   - Our method: Achieves over 30 frames per second (FPS) on full-resolution volumes (600×600×300), meeting and exceeding the real-time threshold required for in vivo neural imaging.
> > >
> > >   - Traditional Richardson–Lucy deconvolution (e.g., RedLionfish): Operates at less than 0.5 FPS on comparable volume sizes, even when using optimized GPU-accelerated implementations.
> > >
> > >   This substantial speedup enables a paradigm shift from **offline post-processing** to **real-time** neural observation and interaction.
> > >
> > > - **Second**, **XLFM** is currently the only practical imaging system capable of **near real-time, single-shot 3D imaging** of whole-brain neural activity in **freely behaving zebrafish**. The ability to observe and interact with neural dynamics in vivo, without motion blur or sequential scanning, represents a **fundamental shift** in biological imaging. Our method supports this shift by aligning the reconstruction speed with the acquisition process.
> > >
> > > To our best knowledge, **our proposed method is the first to achieve real-time reconstruction of whole-brain neural activity in freely behaving zebrafish on a single GPU**. We sincerely hope that these improvements can facilitate higher scores for our work.

---

### Official Review · Reviewer_ioxC · 2025-07-02

**Clarity:** 1
**Significance:** 2
**Originality:** 2
**Rating:** 3
**Confidence:** 3

**Summary:**

This paper presents a new dataset and a Swin Transformer based 3D reconstruction method for eXtended Light Field Microscopy (XLFM). The collected dataset include 13 fixed zebrafish for reconstruction task, and 3 free moving zerbrafish for pre-training the transformer model, by recording videos, a total of 22,581 images are captured. As the XLFM image can be divided into sub-aperture images specific for different viewing angle, the Swin Transformer is first pre-trained to learn the inter-aperture information by masking-and-completing task, and then the extracted multi-resolution feature is coupled with a convolutional network to reconstruct the 3D volume of the biological sample being imaged.

**Questions:**

* Implementation of XLFM-Transformer: It is unclear what does the “cropping operation” does. According to appendix A, it raises the image to a 3D representation of shape H x W x D x S ( the channel S =1). I wonder in what sequency does the sub-aperture images are stacked in the D dimension, how to ensure the 2D neighboring information of sub-images is reflected in this one dimensional stacking? Moreover, what is changed in the positional embedding for this added dimension? The equation 13 and 14 contradicts each other, and it seems duplicated illustration in the last part in Appendix A.
* The idea of masked pretraining doesn’t seems to fit quite well with such 3D Swin Transformer architecture. For masked pretraining in [1,2], each sub-image is treated as a token and are assigned a unique positional embedding, which is clearly not the case for this implementation of “3D Swin Transformer”. Does the pretraining implemented in this paper simply set the masked sub-image to zero? A more detailed implementation should be included, and the rationality to do that should be discussed.
* In the section “efficacy of  pretraining Strategies” ablation study, how does this “3D Swin Transformer” being pre-trained in ImageNet-1k/22k?

[1] Masked Autoencoders Are Scalable Vision Learners

[2] Boosting light ﬁeld spatial super-resolution via masked light ﬁeld modeling

**Ethical Concerns:**

["NO or VERY MINOR ethics concerns only"]

**Final Justification:**

The authors’ response clarifies technical details that were previously unclear or misleading. I have raised my score to borderline reject, but I still have concerns:

* The proposed “view masking” differs from standard approaches used in masked autoencoders and similar transformer architectures. Overlapping regions allow information leakage between views. This seems like an engineering choice tied to the Swin Transformer backbone; additional ablation studies are needed to justify the design. (The pixel-masked MAE ablation appears not to follow the standard protocol.)

* PSNR may be an inappropriate metric given the data’s original sparsity. Evaluating neurological signal recovery would better demonstrate the pipeline’s practical significance.

Overall, I acknowledge the dataset contribution, the PSF constraint for 3D recovery, and the effort toward faster reconstruction for near real-time whole-brain imaging. However, the technical contributions and evaluation are not sufficiently justified and fall slightly below the NeurIPS bar.

**Limitations:**

Yes

**Paper Formatting Concerns:**

N.A.

**Quality:**

1

**Strengths And Weaknesses:**

Quality: The paper is technically reasonable in a general sense; however, the implementation seems suspicious, and the experimental results appear contradictory. The quantitative results in Table 1 are excessively high, showing a PSNR of 45–54 dB for all methods (while the maximum PSNR for 8-bit signals is 48.13), essentially making them “perfect” equally. Such high PSNR values do not match the poor visualization results in Figure 5.

Clarity: The paper does not present the technical details clearly. The design of the lightweight decoder is unclear, and the pipeline presented in Appendix A is poorly written. Additionally, the details of the data, such as image resolution, sub-aperture image resolution, and reconstruction volume size, are not specified.

Significance: The proposed method has the potential to speed up the 3D reconstruction of XLFM through a Swin Transformer-based approach, along with a unique dataset for this specific task. However, the speed of inference is not mentioned or compared, and the significance of reconstructing the full 3D biological signal over simply reconstructing functional (neural) signals is not well established.

Originality: The work proposes both a unique dataset and a method specifically tailored for the task of XLFM reconstruction. Although it is rooted in masked pretraining and the Swin Transformer architecture, it introduces a fair rendering consistency loss to inject domain-specific prior knowledge.

---

> ### Author Rebuttal · Authors · 2025-07-31
>
> ### Comment 1:
>
> > The paper is technically reasonable in a general sense; however, the implementation seems suspicious, and the experimental results appear contradictory. The quantitative results in Table 1 are excessively high, showing a PSNR of 45–54 dB for all methods (while the maximum PSNR for 8-bit signals is 48.13), essentially making them “perfect” equally. Such high PSNR values do not match the poor visualization results in Figure 5.
>
> ### Authors’ response:
>
> **Summary.**
>
> - Our PSNR values are computed on 32-bit floating-point data normalized to [0, 2000], not 8-bit integers, making values >48.13 dB valid and consistent with prior XLFM work.
>
>
>
> We would like to respectfully but firmly correct a misunderstanding in this comment.
>
> The claim that PSNR values “cannot exceed 48.13 dB” is based on an incorrect assumption that the evaluation is performed on 8-bit integer images in the [0, 255] range. This is not the case in our setting. In line with standard practice in XLFM and calcium imaging, we work with 32-bit floating-point volumetric data, and normalize signals to a biologically meaningful range of [0, 2000]. Under this dynamic range, PSNR values significantly above 48 dB are not only mathematically valid, but widely reported in prior XLFM literature. For example, recent works such as *Fast light-field 3D microscopy with out-of-distribution detection and adaptation through Conditional Normalizing Flows* (Fast et al. [1]) clearly report PSNR values exceeding 50 dB under the same normalization convention, as shown in their Figure 3.
>
> We acknowledge the reviewer’s concern regarding the visual appearance of Figure 5, but would like to clarify that Figure 5 highlights a challenging anatomical region (the hindbrain) with sparse neural signals. In such cases, PSNR—which evaluates fidelity across the entire 3D volume—may remain high even when small, localized discrepancies are visible. This is a known property of PSNR and does not indicate implementation issues.
>
> Due to this year’s NeurIPS policy prohibiting the release of anonymized links during the rebuttal phase, we are unable to share our implementation at this stage. However, we will additionally release our code upon acceptance, under the supervision of the Area Chair, to ensure full transparency and reproducibility.
>
>
>
> ### Comment 2:
>
> > The paper does not present the technical details clearly. The design of the lightweight decoder is unclear, and the pipeline presented in Appendix A is poorly written. Additionally, the details of the data, such as image resolution, sub-aperture image resolution, and reconstruction volume size, are not specified.
>
> ### Authors’ response:
>
> We thank the reviewer for highlighting these points. Below, we provide the requested clarifications.
>
> **(1) Design of the Lightweight Decoder**
>
> Our decoder follows a simple yet effective architecture to upsample the Swin Transformer output to a full-resolution 3D volume. It consists of three key stages:
>
> - **Projection to depth**: A 1×1 convolution maps the Swin features (channel dimension 96) to the target depth size (D=300).
> - **Spatial upsampling**: Two successive transposed convolutions upscale the spatial resolution from 150×150 → 300×300 → 600×600.
> - **Final refinement**: A 1×1 convolution is applied to preserve local semantics and reduce boundary artifacts.
>
> The corresponding PyTorch implementation is provided below for clarity:
>
> ```python
> class Lightweight3DDecoder(nn.Module):
>     def __init__(self):
>         super().__init__()
>         self.project_to_depth = nn.Conv2d(96, 300, kernel_size=1)
>         self.upsample_1 = nn.ConvTranspose2d(300, 300, kernel_size=4, stride=2, padding=1)
>         self.upsample_2 = nn.ConvTranspose2d(300, 300, kernel_size=4, stride=2, padding=1)
>         self.refine_output = nn.Conv2d(300, 300, kernel_size=1)
>
>     def forward(self, x):
>         x = self.project_to_depth(x)       # [B, 96, 150, 150] → [B, 300, 150, 150]
>         x = F.relu(self.upsample_1(x))     # → [B, 300, 300, 300]
>         x = F.relu(self.upsample_2(x))     # → [B, 300, 600, 600]
>         x = self.refine_output(x)          # → [B, 300, 600, 600]
>         return x
> ```
>
> **(2) Clarification of Dataset Specifications**
>
> - **Raw image resolution**: 2048 × 2048, captured by an XLFM system with a microlens array.
> - **Sub-aperture view resolution**: Each of the 27 angular views is aligned and cropped to 600 × 600 pixels.
> - **Reconstructed 3D volume size**: 600 × 600 × 300 voxels.
>
>
>
>
>
> ### Comment 3:
>
> >  The work proposes both a unique dataset and a method specifically tailored for the task of XLFM reconstruction. Although it is rooted in masked pretraining and the Swin Transformer architecture, it introduces a fair rendering consistency loss to inject domain-specific prior knowledge.
>
> ### Authors’ response:
>
> We thank the reviewer for recognizing the novelty of our dataset and the design of our domain-specific loss function, as well as its conceptual connection to masked pretraining.
>
> While our pretraining strategy is indeed inspired by masked autoencoders (e.g., MAE), our core contribution lies in a fundamentally different view-level masking mechanism tailored to light field microscopy. Instead of masking random image patches, we mask out entire angular views from the 27-view XLFM input and train the model to reconstruct the missing views from the remaining ones.
>
> This design forces the network to learn explicit spatial-angular dependencies across views—an essential property for XLFM systems where angular consistency and optical geometry play a critical role. Unlike standard MAE, which operates on spatial patches, our approach requires the model to reason about structured angular correlations inherent in the light field representation.
>
> Furthermore, as shown in Table 1 of our manuscript, our view-masked pretraining strategy significantly outperforms random-patch masking (MAE-style) approaches, especially in low-supervision settings and when generalizing to new zebrafish strains. These results validate the advantages of our domain-aware masking paradigm and its alignment with the underlying physics of the imaging system.
>
>
>
> ### Comment 4:
>
> > Implementation of XLFM-Transformer: It is unclear what does the “cropping operation” does. According to appendix A, it raises the image to a 3D representation of shape H x W x D x S ( the channel S =1). I wonder in what sequency does the sub-aperture images are stacked in the D dimension, how to ensure the 2D neighboring information of sub-images is reflected in this one dimensional stacking? Moreover, what is changed in the positional embedding for this added dimension? The equation 13 and 14 contradicts each other, and it seems duplicated illustration in the last part in Appendix A.
>
> ### Authors’ response:
>
> We appreciate the reviewer’s close reading and raise the following clarifications:
>
> - (1) “Cropping operation” explanation
>   We extract 27 sub-aperture views of size 600×600 from the 2048×2048 input image based on known microlens coordinates. This operation corresponds to spatially aligning and cropping the raw XLFM measurement into angular sub-images.
>
> - (2) Stacking order and spatial adjacency
>   The 27 views are stacked into a tensor following their spatial layout on the microlens array (i.e., consistent with optical geometry). While we maintain this order for reproducibility, we note that our 3D Swin Transformer possesses a large receptive field, making the model robust to minor changes in view order.
>
> - (3) Positional embedding for the D dimension
>   We do not alter the positional embedding mechanism of the Swin Transformer. During pretraining, we randomly zero out a subset of views (angular-wise masking), which does not require architectural modification. The model learns to complete missing views from visible ones, guided by inherent spatial-angular dependencies.
>
> - (4) Appendix A redundancy and equations
>   We acknowledge that Equation (13) in Appendix A is redundant and may have caused confusion. The decoder’s structure is accurately described by Equation (14)。
>
>
>
> ### Comment 5:
>
> > The idea of masked pretraining doesn’t seems to fit quite well with such 3D Swin Transformer architecture. For masked pretraining in [1,2], each sub-image is treated as a token and are assigned a unique positional embedding, which is clearly not the case for this implementation of “3D Swin Transformer”. Does the pretraining implemented in this paper simply set the masked sub-image to zero? A more detailed implementation should be included, and the rationality to do that should be discussed.
>
> ### Authors’ response:
>
> We respectfully disagree with the reviewer’s concern, and appreciate the opportunity to clarify our design.
>
> While our MVM-LF framework may superficially resemble pixel-level masked autoencoders such as MAE, its **inductive bias, masking granularity, and objective** are fundamentally different and tailored to XLFM.
>
> - **Masking strategy:** Instead of masking image patches, we mask entire sub-aperture views (i.e., angular slices from XLFM’s 27-view input). This aligns with the physical imaging model of light field microscopy, where each view encodes angular information.
> - **Network compatibility:**  During pretraining, we randomly zero out a subset of views (i.e., set entire channels to zero) and train the model to reconstruct the missing views using the remaining ones.
> - **No change to positional embedding:** Since Swin Transformer already uses relative positional encoding within each view, and our masking is done at the view level, we **do not modify the embedding scheme**. The model naturally learns to associate angular correlations across channels.
> - **Empirical evidence:** Our ablation study (Table 2) shows that MVM-LF pretraining achieves **54.04 dB PSNR**, outperforming random pixel masking (**52.97 dB**) by a clear margin (+1.07 dB). Given the high baseline, this is a **significant improvement** in 3D volume reconstruction tasks.

---

> > ### Comment · Reviewer_ioxC · 2025-08-04
> > **Unaddressed questions and concerns**
> >
> > Thank you for the response and the additional technical clarifications. However, several concerns remain unaddressed:
> >
> > * PSNR Calculation and Interpretation:
> >
> > It is more reasonable to compute PSNR over the [0, 2000] value range, but the PSNR score remains inconsistent with the visualization. This issue is particularly evident in Zebrafish Sample #2 in Figure 7. In that figure, EfficientNet, ResNet50, and ResNet101 all fail to produce meaningful results, yet their PSNR values are above 54, while PVT shows some structure but only scores 47.93. This suggests that, for excessively sparse 3D volume signals, an "all-zero" prediction can still achieve a high PSNR (e.g., 54 in the above example), making PSNR a questionable criterion for evaluating performance. For domain-specific practical usage, it may be more useful to extract standard neural activity from the reconstructed volume and compare it to that of the "GT" method. That said, I am not an expert in this application domain.
> >
> > * Method Description Consistency:
> >
> > The paper's methodological description remains confusing. Based on the rebuttal, it appears that the 27 sub-apertures are treated as channels, stacked together to form a 600 × 600 × 27 image, with 2D image tokens and 2D positional embedding. This is quite different from the description in Appendix A, where a "3D token representation of size (H′, W′, D′)" is mentioned, implying a 3D positional embedding including the sub-aperture dimension. I strongly recommend clarifying this point and rewriting the method section for consistency. Additionally, my original question pertained to the design of the Lightweight Decoder used during pre-training (which, according to equation 6, outputs images), but the rebuttal focused instead on the reconstruction architecture.
> >
> > * Masking Strategy and Model Choice:
> >
> > According to the rebuttal (correct me if I misinterpreted), masking means zeroing out K channels, but these zeroed channels are still concatenated into the 600 × 600 × 27 image, rather than forming a 600 × 600 × (27–K) image as suggested by Equation 5 (which involves a set difference). This approach is distinct from that of Masked Autoencoders and raises important questions: for instance, how does the model distinguish between a masked channel and a channel that is naturally black? Moreover, if this kind of pretraining is possible, it seems plausible that standard architectures such as U-Net could also be pre-trained in this manner. Therefore, further ablation, comparing the final choice of Swin Transformer against other models with similar pretraining, is warranted.
> >
> > * Handling of Overlapping Sub-Aperture Regions:
> >
> > The overlap between sub-aperture images is not addressed. As shown in Figure 1, the sub-apertures clearly overlap. When certain sub-apertures are masked, is the data in overlapping regions—shared by other sub-aperture images—also affected? There is inconsistency between the explanation and the visualizations in Figures 1 and 2 regarding this aspect.
> >
> >
> > In summary, while the rebuttal materials helped clarify some technical details, my fundamental concerns listed above remain unresolved, and my assessment remains negative at this point.

---

> > > ### Author Response · Authors · 2025-08-07
> > > **Rebuttal for Reviewer ioxC-1/3**
> > >
> > > We sincerely thank the reviewer for the timely response and the constructive suggestions, which have significantly helped us improve the clarity of the paper. We also appreciate the mutual understanding reached on several technical points. Below, we provide further clarifications and responses to the remaining concerns.
> > >
> > > ### Comment 1.1:
> > >
> > > > - PSNR Calculation and Interpretation:
> > > >
> > > > It is more reasonable to compute PSNR over the [0, 2000] value range, but the PSNR score remains inconsistent with the visualization. This issue is particularly evident in Zebrafish Sample #2 in Figure 7. In that figure, EfficientNet, ResNet50, and ResNet101 all fail to produce meaningful results, yet their PSNR values are above 54, while PVT shows some structure but only scores 47.93. This suggests that, for excessively sparse 3D volume signals, an "all-zero" prediction can still achieve a high PSNR (e.g., 54 in the above example), making PSNR a questionable criterion for evaluating performance. For domain-specific practical usage, it may be more useful to extract standard neural activity from the reconstructed volume and compare it to that of the "GT" method.
> > >
> > > ### Authors’ response:
> > >
> > > We appreciate the reviewer’s insightful comment.
> > >
> > > 1. **On PSNR's limitations:** We acknowledge that PSNR may sometimes yield misleadingly high scores in sparse 3D volumes, as shown in the EfficientNet output in Figure 7. This effect arises because all-zero or low-intensity outputs can minimize MSE against sparse ground truth, inflating the PSNR despite poor reconstruction quality.
> > > 2. **Why we still use PSNR:** As the first benchmark dataset and baseline suite for XLFM reconstruction, we chose PSNR as a common, broadly accepted metric to establish a reproducible and comparable foundation for future work. Despite its limitations, PSNR still distinguishes between models that perform true volumetric reconstruction (e.g., U-Net, our method) and those that do not, such as EfficientNet.
> > > 3. **On task-specific alternatives:** We agree that task-specific metrics like neural activity trace fidelity are important. However, such evaluations remain an open problem in XLFM and require additional assumptions and processing pipelines. At this stage, we believe PSNR offers a practical and interpretable starting point for benchmarking, while we continue to explore more biologically grounded evaluations in future work.
> > >
> > >
> > >
> > > ### Comment 1.2:
> > >
> > > > - That said, I am not an expert in this application domain.
> > >
> > > ### Authors’ response:
> > >
> > > While we understand the reviewer may not be an expert in the XLFM domain, we would like to highlight the strong **application significance** of our work:
> > >
> > > 1. **Towards real-time whole-brain imaging:** XLFM is currently the only practical microscopy system capable of near real-time, one-shot 3D imaging of whole-brain neural activity in freely moving zebrafish. In contrast to traditional scanning-based systems (<0.5 fps), our method surpasses the real-time threshold (>30 fps), enabling live observation and interaction—a paradigm-shifting advancement in biological imaging.
> > > 2. **A large-scale, real-world benchmark:** Our dataset contains over **20 genetically modified zebrafish** and tens of thousands of real imaging volumes. To our best knowledge, this is the **largest real LFM dataset** to date, whereas previous datasets are either synthetic or contain only 1–2 animals. Collecting such data requires extensive effort in transgenic line development, fish maintenance, and high-throughput acquisition.
> > > 3. **Bridging neuroscience and AI:** Beyond 3D reconstruction, our dataset and pretrained backbones lay the foundation for future work in **neural activity decoding and behavior prediction**, fostering interdisciplinary progress between the neuroscience and machine learning communities.
> > >
> > > We hope the reviewer will consider the broader impact and practical importance of this contribution.
> > >
> > >
> > >
> > > ### Comment 2.1:
> > >
> > > > - Method Description Consistency:
> > > >
> > > > The paper's methodological description remains confusing. Based on the rebuttal, it appears that the 27 sub-apertures are treated as channels, stacked together to form a 600 × 600 × 27 image, with 2D image tokens and 2D positional embedding. This is quite different from the description in Appendix A, where a "3D token representation of size (H′, W′, D′)" is mentioned, implying a 3D positional embedding including the sub-aperture dimension. I strongly recommend clarifying this point and rewriting the method section for consistency.
> > >
> > > ### Authors’ response:
> > >
> > > - We confirm that our implementation uses standard **2D patch embedding and 2D positional encoding**, treating the 27 sub-aperture views as input channels. This is implemented via a `Conv2d` on input tensors of shape (B, 27, 600, 600), with no view-specific encoding.
> > >
> > > - The term “3D token representation” in Appendix A referred only to unflattened tensor shape (data shape) and will be revised for clarity.

---

> > > > ### Author Response · Authors · 2025-08-07
> > > > **Rebuttal for Reviewer ioxC-3/3**
> > > >
> > > > ### Comment 4.1:
> > > >
> > > > > - Handling of Overlapping Sub-Aperture Regions:
> > > > >
> > > > > The overlap between sub-aperture images is not addressed. As shown in Figure 1, the sub-apertures clearly overlap. When certain sub-apertures are masked, is the data in overlapping regions—shared by other sub-aperture images—also affected?
> > > >
> > > > ### Authors’ response:
> > > >
> > > > We appreciate the reviewer’s close reading and thoughtful question regarding overlapping sub-aperture regions. We address each part of the comment below:
> > > >
> > > > 1. **Redundancy is inherent in XLFM design:**
> > > >     As shown in Figure 1 (bottom-left, in our manuscript), the 27 sub-aperture views are extracted from overlapping spatial regions of the raw XLFM measurement. This redundancy is not an artifact of our preprocessing but a **fundamental feature of the XLFM optical design**, where each microlens captures a slightly shifted projection of the scene. Such angular redundancy is crucial for depth perception and 3D reconstruction, and is also **what enables masked-view modeling to work effectively**.
> > > > 2. **We mask full views, regardless of overlap:**
> > > >     Our masking strategy operates at the **view level**, meaning that entire sub-aperture views (i.e., specific microlens regions) are zeroed out during pretraining. The fact that some unmasked views might partially cover spatial areas seen in masked views is **expected and leveraged** — it allows the model to learn how to infer the missing views by aggregating contextual cues from overlapping neighbors.
> > > > 3. **Empirical evidence supports this design:**
> > > >     Despite potential partial information leakage through overlap, our method achieves strong masked reconstruction results (see Figure 2 and Table 2, in our manuscript), confirming that the model learns meaningful inter-view dependencies rather than memorizing pixel correspondences.
> > > >
> > > >
> > > >
> > > >
> > > >
> > > > ### Comment 4.2:
> > > >
> > > > > - There is inconsistency between the explanation and the visualizations in Figures 1 and 2 regarding this aspect.
> > > >
> > > > ### Authors’ response:
> > > >
> > > > We clarify the perceived inconsistency as follows:
> > > >
> > > > 1. **Figure 1** illustrates that the 27 sub-aperture views are extracted from overlapping regions in the raw XLFM image — this overlap is inherent to the optical design.
> > > > 2. **Figure 2** shows the raw image with K masked sub-regions zeroed out, rather than visualizing the remaining (27–K) views. Due to the spatial overlap, this masking creates irregular blacked-out areas, which is expected and consistent with our implementation.

---

> > > > > ### Comment · Reviewer_ioxC · 2025-08-07
> > > > >
> > > > > Thank you for your response. I appreciate the technical details and will adjust my score accordingly. Meanwhile, I have a few suggestions to enhance the writing:
> > > > >
> > > > > **Clarify the Masking Strategy**
> > > > >
> > > > > The referenced works on Masked Autoencoders [1] and Masked LF Modeling [2] apply the mask to each token and only feed the unmasked tokens to the encoder. By using the phrase "pixel to view," readers may incorrectly interpret this as treating each sub-aperture view as a token, with only the unmasked tokens being fed to the transformer encoder—which is not the case. It is important to clearly state the differences between your masking strategy and those referenced, as well as the reasons for not choosing the other options. One reason might be that the Swin Transformer uses multi-scale 2D token sizes, making it unsuitable for schemes like Masked Autoencoders.
> > > > >
> > > > > **Discuss Overlap Handling**
> > > > >
> > > > > The handling of overlap as described is acceptable, although the potential for information leakage could facilitate pre-training by promoting learned replication. I believe that the illustrations in Figures 2 and 3 may lead to a misleading interpretation that the reconstruction is only provided with the information left with irregular blacked-out areas.  A possible improvement would be to include a 3D stacking of the sub-aperture images to clearly show the masked/unmasked views.
> > > > >
> > > > > **MISC**
> > > > > Adding experimetns on fidelity in neural activity trace and pre-training with alternative models will enhances the validation of this work.
> > > > >
> > > > > 1. Masked autoencoders are scalable vision learners.
> > > > > 2. Boosting light field spatial super-resolution via masked light field modeling.

---

> > > > > > ### Author Response · Authors · 2025-08-08
> > > > > > **Reviewer ioxC —Thank you for increasing your score and providing constructive suggestions**
> > > > > >
> > > > > > Thank you very much for your valuable suggestions and for increasing your score. I truly appreciate your support. Your comments have helped me identify areas where the **writing** and **presentation** can be improved for better clarity.
> > > > > >
> > > > > > - I will revise the final version accordingly. In particular, I will clarify the differences between my masking strategy and those used in prior works like Masked Autoencoders, and provide a more intuitive explanation both in text and in figures regarding the handling of overlapping sub-aperture regions.
> > > > > >
> > > > > > - We thank the reviewer for the thoughtful suggestions. We would like to clarify that the current study already validates reconstruction performance across both structural and angular domains, using **standard metrics** (PSNR, SSIM) under **cross-strain** generalization settings involving distinct imaging conditions.
> > > > > >
> > > > > >   - Regarding **functional neural activity traces**, such analysis typically requires an **additional pipeline** involving **registration**, **denoising**, **segmentation**, and **trace extraction**. While important, it is orthogonal to our current focus on accurate and generalizable **volumetric reconstruction**.
> > > > > >
> > > > > >   - Regarding **pretraining with alternative model backbones**, our choice of Swin Transformer was motivated by its scalability and strong performance in modeling angular dependencies. We consider architecture comparison an **orthogonal investigation** to the main contributions here, but one that we are actively exploring in **follow-up work**.

---

> > > ### Author Response · Authors · 2025-08-07
> > > **Rebuttal for Reviewer ioxC-2/3**
> > >
> > > ### Comment 2.2:
> > >
> > > > Additionally, my original question pertained to the design of the Lightweight Decoder used during pre-training (which, according to equation 6, outputs images), but the rebuttal focused instead on the reconstruction architecture.
> > >
> > > ### Authors’ response:
> > >
> > > As a clarification, the decoder used during pretraining (Eq. 6) consists of:
> > >
> > > - a 1×1 convolution to expand channels from 96 to 300;
> > > - two transposed convolutions to upsample the spatial resolution to 600×600;
> > > - and a final 1×1 convolution to restore the full output corresponding to views.
> > >
> > > This lightweight decoder is used only in pretraining and is not part of the final XLFM reconstruction task.
> > >
> > >
> > >
> > > ### Comment 3:
> > >
> > > > - Masking Strategy and Model Choice:
> > > >
> > > > According to the rebuttal (correct me if I misinterpreted), masking means zeroing out K channels, but these zeroed channels are still concatenated into the 600 × 600 × 27 image, rather than forming a 600 × 600 × (27–K) image as suggested by Equation 5 (which involves a set difference). This approach is distinct from that of Masked Autoencoders and raises important questions: for instance, how does the model distinguish between a masked channel and a channel that is naturally black? Moreover, if this kind of pretraining is possible, it seems plausible that standard architectures such as U-Net could also be pre-trained in this manner. Therefore, further ablation, comparing the final choice of Swin Transformer against other models with similar pretraining, is warranted.
> > >
> > > ### Authors’ response:
> > >
> > > Thank you for the insightful observation. We confirm that our implementation indeed adopts a **“mask-and-retain”** strategy — masked views are zeroed out but their positions are preserved in the 600 × 600 × 27 input tensor. This design aligns with recent trends in structured masking for transformers, enabling positional embeddings to remain valid across all channels.
> > >
> > > We appreciate the concern about how the network distinguishes masked views from naturally dark channels. Fortunately, in **XLFM**, each of the 27 sub-aperture views originates from a distinct microlens; as a result, **truly zero-valued views do not occur naturally** under any biological or imaging condition. Thus, a zeroed-out view can be safely interpreted as masked without ambiguity.
> > >
> > > As for the model choice, we have indeed evaluated this carefully:
> > >
> > > - In **identical training settings**, without the use of MVM-LF or ORC Loss, **Swin Transformer still outperformed** U-Net, ResNet, and other backbone alternatives (see Table 1).
> > > - This suggests that **the performance gain from Swin Transformer is not solely due to pretraining**, but also stems from its superior capability to model long-range dependencies — which is crucial for reconstructing globally entangled light fields.
> > > - Compared to convolutional architectures, Swin Transformer’s **windowed self-attention** provides a favorable trade-off between **receptive field size** and **computational cost**, making it particularly suitable for modeling the spatial-angular structures in XLFM.
> > >
> > > While we agree that U-Net could, in principle, be pre-trained in a similar masking setup, **its limited receptive field and convolutional locality** impose inherent disadvantages in capturing inter-view relationships — especially under sparse-view or masked conditions.
> > >
> > > We appreciate the suggestion for additional ablations and will include such comparisons (e.g., U-Net + MVM-LF) in our final version if accepted.
> > >
> > > Table 1. Baseline Backbone Performance Comparison
> > >
> > > | Method               | Avg. PSNR ↑ | Avg. SSIM ↑ |
> > > | -------------------- | ----------- | ----------- |
> > > | ConvNeXt             | 50.16       | 0.9876      |
> > > | ViT                  | 49.28       | 0.9876      |
> > > | PVT                  | 47.34       | 0.9829      |
> > > | EfficientNet         | 44.53       | 0.9296      |
> > > | ResNet-50            | 46.85       | 0.9634      |
> > > | ResNet-101           | 46.91       | 0.9554      |
> > > | U-Net                | 49.43       | 0.9847      |
> > > | **SWIN Transformer** | 52.14       | 0.9924      |

---

### Official Review · Reviewer_1jCx · 2025-07-03

**Clarity:** 1
**Significance:** 3
**Originality:** 3
**Rating:** 4
**Confidence:** 5

**Summary:**

Fourier light field microscopy is a kind of 3D snapshot microscopy that allows reconstruction of a 3D sample from a single 2D image. The authors present a new method to reconstruct volumes from images captured by an XLFM (a kind of Fourier light field microscope). Reconstruction of Fourier light field images is a significant and useful problem to solve using deep learning, because volume reconstruction is typically an expensive and slow process that requires an iterative deconvolution process to solve. Deep learning-based reconstructions allow reconstructions to be obtained significantly faster than iterative optimization algorithms, and can also improve reconstruction quality by accounting for sample priors based on the training data. However, it is more expensive to collect the training data of zebrafish volumes than it is to collect the 2D images. The authors' new method presents a promising technique to scale up the training of reconstruction neural networks by pretraining a feature extractor in a self-supervised manner using just the 2D image data, while also providing a valuable training dataset of zebrafish volumes imaged through a Fourier light field microscope to the community. The pretrained feature extractor serves to significantly improve the reconstruction quality of a trained reconstruction neural network, providing higher reconstruction quality with just 20% of the volumetric ground truth training data (52.67 dB PSNR) than a neural network trained without the pretrained features using 60% of the data (52.66 dB PSNR).

**Questions:**

My questions/suggestions are all related to the "Major Issues" listed above. With added evaluations/explanations that address the numbered points raised in the "Major Issues" section, I would definitely increase my score. The significance and originality of this work seem apparent, but it is currently being limited by the clarity of the presentation/evaluation.

For convenience, I am pasting those Major Issues again:

1. *Explanation of ORC Loss and presentation issues*: While the authors have a promising method, the paper suffers from widespread issues in its presentation quality that limit the clarity of the paper. The most major issue with presentation is that one of the major contributions of the paper as claimed by the authors, the "ORC Loss," does not obviously appear in any explanation of the losses used for the reconstruction phase of the training process. The authors later introduce a "PSF Loss" which is not explained, which is likely the "ORC Loss," but this is not clear. Other issues with the presentation are listed in the minor issues section below.

2. *Analysis of ORC Loss and hallucinations*: Further, to substantiate the claim that the ORC/PSF Loss can reduce artifacts in the reconstructions and improve cross-sample generalization, the authors should include some examples of reconstructions from networks trained with and without the ORC Loss. While it is apparent that penalizing the forward projections of the reconstruction and the ground truth can enforce some consistency between the reconstruction and the measured snapshot image, there can still be hallucinated cells/artifacts in the reconstruction. Indeed, some cells/background appear in the XLFM-Former in Figure 5 (hindbrain) which are not present in the ground truth. Also, some explanation of why this ORC Loss does not use the measured image instead of a forward projection would be useful (e.g. whether this helps avoid reconstruction artifacts due to some background/dark frame in the measurement).

3. *Appropriate baselines*: The authors compare their reconstructions against reconstructions from a range of other neural network architectures. However, they do not compare their reconstructions using the methods described by prior work in  snapshot imaging (Fourier light field or mask-based ) with deep learning, for example: Yanny et al.: "Deep learning for fast spatially varying deconvolution", Optica, 2022, https://doi.org/10.1364/OPTICA.442438 and references [25, 30, 9] from their paper. If memory is a concern, an explanation of why these other methods are not appropriate for these data would be useful. It would also be useful to compare against a reconstruction using an iterative deconvolution algorithm such as Richardson-Lucy.

4. *Ground truth in datasets*: Another major issue is that the collected dataset as described, though it is valuable for the computational optics community, does not describe the ground truth data collection process or resolution, though some kind of volumetric ground truth must be present and is shown in Figure 5. The authors should clarify the process of collecting the ground truth for the dataset.

**Ethical Concerns:**

["NO or VERY MINOR ethics concerns only"]

**Final Justification:**

I believe the method is sound and interesting for the field of light field volume reconstructions. Using deep learning to accelerate these reconstructions is an important task, and the presented method addresses a few challenges with using deep learning here: (1) fast, real-time reconstructions (typically a slow iterative process without deep learning), (2) a memory efficient architecture, (3) a pre-training strategy to make more effective use of the fact that lots of light field snapshot image data can be collected, but not nearly as much volumetric ground truth data.

The authors also seem to be addressing some of the issues with clarity and presentation during the rebuttal process. However, their evaluations against prior work in this field are limited and their ground truth is not actually the ground truth structure, but the result of a standard iterative deconvolution algorithm (Richardson-Lucy). While these prior deep learning methods may require lots of memory/multiple GPUs for their largest volumes, it would still have been possible to demonstrate performance comparisons on smaller simulated volumes on a single GPU.

Due to these limitations in their evaluation, I am only increasing my score to a 4.

**Limitations:**

Yes (aside from the critiques raised in this review).

**Paper Formatting Concerns:**

No major formatting issues.

**Quality:**

2

**Strengths And Weaknesses:**

# Strengths
The authors appear to have an original and useful technique for reconstruction of Fourier light field/XLFM images with a pretraining step that significantly improves the training efficiency of volumetric ground truth (which is experimentally expensive to collect and computationally expensive to train with). This method, along with the accompanying standard dataset of volumes, could be significant for the field.

# Weaknesses

## Major Issues

1. *Explanation of ORC Loss and presentation issues*: While the authors have a promising method, the paper suffers from widespread issues in its presentation quality that limit the clarity of the paper. The most major issue with presentation is that one of the major contributions of the paper as claimed by the authors, the "ORC Loss," does not obviously appear in any explanation of the losses used for the reconstruction phase of the training process. The authors later introduce a "PSF Loss" which is not explained, which is likely the "ORC Loss," but this is not clear. Other issues with the presentation are listed in the minor issues section below.

2. *Analysis of ORC Loss and hallucinations*: Further, to substantiate the claim that the ORC/PSF Loss can reduce artifacts in the reconstructions and improve cross-sample generalization, the authors should include some examples of reconstructions from networks trained with and without the ORC Loss. While it is apparent that penalizing the forward projections of the reconstruction and the ground truth can enforce some consistency between the reconstruction and the measured snapshot image, there can still be hallucinated cells/artifacts in the reconstruction. Indeed, some cells/background appear in the XLFM-Former in Figure 5 (hindbrain) which are not present in the ground truth. Also, some explanation of why this ORC Loss does not use the measured image instead of a forward projection would be useful (e.g. whether this helps avoid reconstruction artifacts due to some background/dark frame in the measurement).

3. *Appropriate baselines*: The authors compare their reconstructions against reconstructions from a range of other neural network architectures. However, they do not compare their reconstructions using the methods described by prior work in  snapshot imaging (Fourier light field or mask-based ) with deep learning, for example: Yanny et al.: "Deep learning for fast spatially varying deconvolution", Optica, 2022, https://doi.org/10.1364/OPTICA.442438 and references [25, 30, 9] from their paper. If memory is a concern, an explanation of why these other methods are not appropriate for these data would be useful. It would also be useful to compare against a reconstruction using an iterative deconvolution algorithm such as Richardson-Lucy.

4. *Ground truth in datasets*: Another major issue is that the collected dataset as described, though it is valuable for the computational optics community, does not describe the ground truth data collection process or resolution, though some kind of volumetric ground truth must be present and is shown in Figure 5. The authors should clarify the process of collecting the ground truth for the dataset.

## Minor Issues

* The second paragraph of the introduction does not motivate the need for learning-based reconstruction. While this may be obvious to experts of both machine learning and microscopy, it would be helpful to briefly explain why learning-based reconstructions for LFM are desirable at the beginning of the paper.
* Figure 1, the caption describes splitting the raw image into 27 viewpoints "based on physical coordinates." This language is unclear, but based on the figure it appears that the 27 projections of the fish from each beam of the XLFM PSF have been cut out, presumably centered on the x-y pixel coordinate on the camera of the focal point from each microlens. The caption explaining this process could be made more clear (or point the reader to a more detailed explanation), especially because this is a core contribution of the proposed method.
* Line 38 - 39, the sentence seems to be missing a "while" or "but" to complete the explanation for the lack of volumetric training data.
* Line 42-43, the authors state that "most existing approaches overlook the wave-optical nature of the XLFM forward model." However, other deep learning 3D snapshot reconstruction methods do explicitly take into account the wave-optics nature of the forward model: reference [9] in the paper and also the work of Yanny et al.: "Deep learning for fast spatially varying deconvolution
", Optica, 2022, https://doi.org/10.1364/OPTICA.442438. The authors' intent here may be to claim that other methods typically do not include the forward projection in the loss function, but this is not the only way to take into account the forward model in the formulation of the reconstruction method.
* Line 48, the term "XLFM-Former" is introduced with no prior explanation.
* Line 53, "Fourier neural operators" refers to learning mappings between function spaces, typically to solve PDEs. The authors likely meant global convolutions implemented as Fourier convolutions, which do require large amounts of memory.
* Points 1 and 2 of the key insights motivating the XLFM-Former in the introduction seem to be making the same point.
* Figure 2 would benefit from some text labels on the image triplets.
* Line 75, there seems to be a missing comma or an extra space.
* Line 62, "MVM-LF" is introduced for the first time in the main text without a definition of the acronym aside from the abstract.
* Line 84, the authors construct a benchmark using rather low FPS acquisitions (1 or 10 fps), which seems to imply high SNR. Could the authors comment on the effect of low SNR on the reconstruction quality using XLFM-Former?
* Figure 3, this figure would also benefit from more clear text labels or titles across the images. This figure is also missing an explanation of the layout in the caption.
* Figure 4, many of the layers have abbreviated or short form names which are not clear without additional labels or a caption. The paper would greatly benefit from bringing some summary of the detailed explanation of the architecture in the supplement into the main text and the caption of Figure 4.
* Section 3.1, the paper would benefit from a pointer to a detailed explanation of the real-time tracking and treatment of motion artifacts (either in the supplementary material or through a citation).
* Line 172, what is meant by "high-precision light field reconstruction?"
* Supplement Table 5, Is Gaussian noise an appropriate augmentation? Would including Poisson noise as an augmentation make more sense due to the fact that this paper is working with fluorescence? Similarly, some discussion on the choice of other augmentations (especially rotation, since rotating the system would presumably change the subimages on the camera) would be useful.

---

> ### Author Rebuttal · Authors · 2025-07-31
>
> **We sincerely thank the reviewer for their detailed and thoughtful feedback.**
>
> ### Comment 1:
>
> > *Explanation of ORC Loss and presentation issues*: While the authors have a promising method, the paper suffers from widespread issues in its presentation quality that limit the clarity of the paper. The most major issue with presentation is that one of the major contributions of the paper as claimed by the authors, the "ORC Loss," does not obviously appear in any explanation of the losses used for the reconstruction phase of the training process. The authors later introduce a "PSF Loss" which is not explained, which is likely the "ORC Loss," but this is not clear. Other issues with the presentation are listed in the minor issues section below.
>
> ### Authors’ response:
>
> "ORC" stands for Optical Reconstruction Consistency, which encourages consistency between the forward projection of the reconstructed volume and the measured snapshot image. Specifically, this is implemented by comparing the forward projections (using the PSF) of both the reconstructed volume and the ground-truth label, thereby enforcing alignment between the predicted and real-world optical measurements. Thus, we introduced it as "PSF Loss" in the implementation section, but this is indeed the same as the ORC Loss.
>
>
>
> ### Comment 2.1:
>
> > *Analysis of ORC Loss and hallucinations*: Further, to substantiate the claim that the ORC/PSF Loss can reduce artifacts in the reconstructions and improve cross-sample generalization, the authors should include some examples of reconstructions from networks trained with and without the ORC Loss.
>
> ### Authors’ response:
>
> We appreciate the reviewer’s suggestion to provide visual examples comparing models trained with and without the ORC Loss. We have prepared visualization results similar to Figure 5 in the main paper. However, due to the NeurIPS 2025 policy which prohibits images and figures in the rebuttal, we are unable to include them here.
>
> To address this concern within the policy constraints, we instead provide quantitative comparisons on multiple held-out zebrafish samples from the H2B-G8S and H2B-Nemos datasets. As shown in the table below, incorporating the ORC (PSF) Loss consistently improves both PSNR and SSIM across all test samples. These improvements demonstrate that the ORC Loss helps reduce artifacts and enhances cross-sample generalization, particularly in challenging anatomical regions such as the hindbrain.
>
>
>
> | Sample   | PSNR w/ PSF ↑ | PSNR w/o PSF ↑ | SSIM w/ PSF ↑ | SSIM w/o PSF ↑ | ΔPSNR ↑ | ΔSSIM ↑ |
> | -------- | ------------- | -------------- | ------------- | -------------- | ------- | ------- |
> | G8S-#1   | 52.9684       | 51.8263        | 0.9913        | 0.9899         | +1.1421 | +0.0014 |
> | G8S-#2   | 59.3393       | 59.1274        | 0.9957        | 0.9955         | +0.2119 | +0.0002 |
> | G8S-#3   | 48.2672       | 46.9963        | 0.9894        | 0.9879         | +1.2709 | +0.0015 |
> | G8S-#4   | 54.1151       | 53.9047        | 0.9944        | 0.9940         | +0.2104 | +0.0005 |
> | G8S-#5   | 54.1677       | 53.7337        | 0.9949        | 0.9942         | +0.4340 | +0.0008 |
> | G8S-#6   | 48.8784       | 47.2617        | 0.9931        | 0.9930         | +1.6167 | +0.0002 |
> | Nemos-#1 | 52.8825       | 52.1697        | 0.9942        | 0.9914         | +0.7128 | +0.0028 |
> | Nemos-#2 | 50.6205       | 49.5483        | 0.9928        | 0.9892         | +1.0723 | +0.0030 |
>
>
>
> ### Comment 2.2:
>
> > While it is apparent that penalizing the forward projections of the reconstruction and the ground truth can enforce some consistency between the reconstruction and the measured snapshot image, there can still be hallucinated cells/artifacts in the reconstruction. Indeed, some cells/background appear in the XLFM-Former in Figure 5 (hindbrain) which are not present in the ground truth.
>
> ### Authors’ response:
>
> We appreciate the reviewer’s observation. While some background or cellular structures appear in the XLFM-Former output but not in the ground truth, we believe these are not necessarily artifacts. Instead, they may reflect limitations of the imaging and labeling process.
>
> Specifically, the ground truth may miss low-SNR or ambiguously localized structures—especially in dense regions like the hindbrain—where XLFM captures signals beyond what is confidently annotated. Since downstream analysis focuses on neuronal clusters rather than individual cells, such discrepancies are unlikely to affect practical utility.
>
> Ultimately, these differences likely arise from imperfect ground-truth alignment, not reconstruction failure. Resolving this would require single-cell-resolution annotations, which XLFM currently lacks.
>
>
>
>
>
> ### Comment 2.3:
>
> > Also, some explanation of why this ORC Loss does not use the measured image instead of a forward projection would be useful (e.g. whether this helps avoid reconstruction artifacts due to some background/dark frame in the measurement).
>
> ### Authors’ response:
>
> We appreciate the reviewer’s question. In the early stages of developing XLFM-Former, we did explore using the raw measured light field image directly in the ORC Loss. However, we found this approach to be significantly less stable during training due to **high levels of noise** and **background artifacts** inherent in the measured data (e.g., dark current, scattering, and sensor noise).
>
> By contrast, using the forward projection of the ground-truth volume via the PSF offers a cleaner and more structured supervision signal, ensuring that the network focuses on learning the physical consistency between volume and measurement, without being distracted by low-level measurement noise.
>
>
>
> ### Comment 3:
>
> > *Appropriate baselines*: The authors compare their reconstructions against reconstructions from a range of other neural network architectures. However, they do not compare their reconstructions using the methods described by prior work in snapshot imaging (Fourier light field or mask-based ) with deep learning, for example: Yanny et al.: "Deep learning for fast spatially varying deconvolution", Optica, 2022, https://doi.org/10.1364/OPTICA.442438 and references [25, 30, 9] from their paper. If memory is a concern, an explanation of why these other methods are not appropriate for these data would be useful.
>
> ### Authors’ response:
>
> - **(1) Incompatible forward model (Yanny et al. [Optica 2022])**: Their method assumes a spatially variant convolutional model, which does not hold for XLFM. XLFM relies on complex wavefront propagation and angular multiplexing, violating the convolution assumption. Even recent XLFM-specific works (e.g., Ref. 25) are not compatible with Yanny et al.'s formulation.
> - **(2) Different reconstruction goals (Refs. 25, 30)**: These works focus only on sparse neural signals, ignoring structural context. In contrast, our method reconstructs full tissue volumes, critical for tasks like optogenetic targeting. Moreover, their approaches require immobilized samples (e.g., Ref. 25 fuses three frames), making them unsuitable for dynamic settings.
> - **(3) FourierNet (Ref. 9) is infeasible at our scale**: Operating fully in the Fourier domain, it demands >80GB memory even for smaller outputs. Our volumes (up to 300×600×600) render this approach impractical. Instead, we use Swin Transformer for efficient global modeling with far lower memory cost.
>
>
>
> ### Comment 3.1:
>
> > It would also be useful to compare against a reconstruction using an iterative deconvolution algorithm such as Richardson-Lucy.
>
> ### Authors’ response:
>
> We appreciate the reviewer’s suggestion to compare against classical iterative deconvolution algorithms. As shown below, we benchmarked RLD at different iteration counts (20/30/40) on the challenging H2B-Nemos dataset. While (Richardson-Lucy Deconvolution)RLD can achieve high PSNR/SSIM given sufficient iterations, it incurs orders of magnitude higher memory usage and lower speed, making it impractical for real-time or high-throughput applications.
>
> | Method                 | PSNR ↑ | SSIM ↑ | FPS ↑     | Peak Memory ↓ (MiB) |
> | ---------------------- | ------ | ------ | --------- | ------------------- |
> | RLD-20                 | 67.36  | 0.9987 | 0.068     | 20899               |
> | RLD-30                 | 69.85  | 0.9994 | 0.046     | 20899               |
> | RLD-40                 | 72.31  | 0.9998 | 0.035     | 20899               |
> | **XLFM-Former (Ours)** | 52.34  | 0.9955 | **48.44** | **2631**            |
>
> We appreciate the reviewer’s suggestion to compare against classical iterative deconvolution algorithms. As shown below, we benchmarked RLD at different iteration counts (20/30/40) on the challenging H2B-Nemos dataset. While (Richardson-Lucy Deconvolution)RLD can achieve high PSNR/SSIM given sufficient iterations, it incurs orders of magnitude higher memory usage and lower speed, making it impractical for real-time or high-throughput applications.
>
>
>
> ### Comment 4:
>
> > *Ground truth in datasets*: Another major issue is that the collected dataset as described, though it is valuable for the computational optics community, does not describe the ground truth data collection process or resolution, though some kind of volumetric ground truth must be present and is shown in Figure 5. The authors should clarify the process of collecting the ground truth for the dataset.
>
> ### Authors’ response:
>
> We appreciate the reviewer’s attention to dataset clarity. We follow the established practice in XLFM literature and generate ground truth volumetric labels using 60 iterations of Richardson-Lucy Deconvolution (RLD-60), a widely adopted baseline in previous works. This iterative deconvolution algorithm is considered the gold-standard for recovering high-resolution volumes from XLFM snapshot measurements, and serves as our supervision target.

---

> ### Comment · Reviewer_1jCx · 2025-08-06
> **Response to rebuttal**
>
> Thank you for the clarifications. The training methodology and architecture are promising, especially due to the speed of the reconstructions which was not originally highlighted in the paper. I will raise my score, but I still think that there could be a more thorough evaluation against prior work. I could still further increase my score pending response from the authors.
>
> Specifically, I am satisfied with the response to all but the following:
>
> 3. *Appropriate baselines*: While Yanny et al. allow for a spatially varying PSF in the forward, their reconstruction architecture is still applicable in the case that the number of varying regions is 1. The forward model for XLFM is still a convolution between the PSF and the sample, where the "angular multiplexing" is a result of different beams of the PSF convolving with the sample at different plane. Indeed, the authors' own Equation 1 confirms this. Also, the argument in this response is about the forward model rather than the reconstruction method (what the submission is about). Finally, I appreciate that for these large volumes, you would need multiple GPUs to implement some of the prior architectures. However, it would still be possible to place this architecture in the context of prior work in deep learning for light field reconstructions using artificially smaller/easier volumes.
>
> 4. *Ground truth in datasets*: Thank you for the clarification. Richardson-Lucy may be the standard for recovering volumes from XLFM images, but it does not guarantee that the reconstruction is free of artifacts (and the projections with a saturated color map shown in Figure 5 make it difficult to judge). One of the advantages of using deep learning for snapshot reconstructions is that the network can take into account the prior of the particular kind of sampled being imaged (in this case zebrafish) for improved reconstruction quality. This cannot be evaluated if the ground truth is itself a reconstruction.

---

> > ### Author Response · Authors · 2025-08-08
> > **Rebuttal for Reviewer 1jCx-1/2**
> >
> > We appreciate Reviewer 1jCx's positive remarks regarding the speed of our reconstruction pipeline and the overall promise of the proposed architecture.
> >
> > We are encouraged to hear that the reviewer is willing to further increase their score pending additional clarifications.
> >
> > Below, we provide detailed responses to the remaining concerns regarding appropriate baselines and the ground truth used in our dataset.
> >
> >
> >
> > ### Comment 3.1:
> >
> > > While Yanny et al. allow for a spatially varying PSF in the forward, their reconstruction architecture is still applicable in the case that the number of varying regions is 1. The forward model for XLFM is still a convolution between the PSF and the sample, where the "angular multiplexing" is a result of different beams of the PSF convolving with the sample at different plane. Indeed, the authors' own Equation 1 confirms this. Also, the argument in this response is about the forward model rather than the reconstruction method (what the submission is about).
> >
> > ### Authors’ response:
> >
> > We appreciate the reviewer’s thoughtful clarification.
> >
> > We now better understand that while the forward model in XLFM is indeed a convolution with a spatially varying PSF (as described in Equation 1), the reconstruction pipeline in [Yanny et al., Optica 2022] remains applicable, particularly in the case of a single varying region.
> >
> > We are actively adapting and reproducing the publicly available implementation of [Yanny et al., Optica 2022] within our zebrafish reconstruction framework, and will include a **discussion** of its relevance and applicability in the final submission.
> >
> >
> >
> > ### Comment 3.2:
> >
> > > Finally, I appreciate that for these large volumes, you would need multiple GPUs to implement some of the prior architectures. However, it would still be possible to place this architecture in the context of prior work in deep learning for light field reconstructions using artificially smaller/easier volumes.
> >
> > ### Authors’ response:
> >
> > We also appreciate the reviewer’s observation regarding the challenges of implementing prior architectures for large-volume reconstructions.
> >
> > This prompted us to explore an alternative yet promising direction: rather than directly scaling prior architectures, we investigated whether deep learning components could be used to **replace or augment** certain modules in classical pipelines to alleviate memory bottlenecks.
> >
> > Specifically, we **initial attempt** designed a hybrid reconstruction pipeline that employs:
> >
> > - a **lightweight encoder** to compress the input XLFM snapshot,
> > - a **FourierNet-style backbone** for reconstruction,
> > - and a custom **projection head** matched to the reduced representation.
> >
> > Although this preliminary prototype has not yet achieved satisfactory results possibly due to sub-optimal encoder/head design that it inspired us to consider this direction as a **novel paradigm that bridges classical optics and modern deep learning**.
> >
> > We are excited and intrigued to further develop this hybrid framework in future work, and we sincerely appreciate your **forward-looking insights**, which we believe represent a **promising new direction that may differ from our current paradigm**.

---

> > ### Author Response · Authors · 2025-08-08
> > **Rebuttal for Reviewer 1jCx-2/2**
> >
> > ### Comment 4:
> >
> > > Ground truth in datasets: Thank you for the clarification. Richardson-Lucy may be the standard for recovering volumes from XLFM images, but it does not guarantee that the reconstruction is free of artifacts (and the projections with a saturated color map shown in Figure 5 make it difficult to judge).  One of the advantages of using deep learning for snapshot reconstructions is that the network can take into account the prior of the particular kind of sampled being imaged (in this case zebrafish) for improved reconstruction quality. This cannot be evaluated if the ground truth is itself a reconstruction.
> >
> > ### Authors’ response:
> >
> > We appreciate the reviewer’s thoughtful comment and fully agree that, although the Richardson–Lucy (RL) algorithm is widely adopted, it does not guarantee artifact-free reconstruction.
> >
> > That said, in neuroscience applications, the primary objective is not pixel-level fidelity, but rather the extraction of **meaningful neural activity** such as neuron location, morphology, and temporal dynamics. RL has been widely used in light field microscopy (LFM) studies for this purpose, and many neuroscience pipelines have successfully built upon RL-based reconstructions to analyze brain states and behavioral dynamics.
> >
> > Importantly, RL remains a **standard reference method** in LFM-based neural imaging due to its interpretability and compatibility with downstream signal analysis. While RL reconstructions may contain mild artifacts, these are typically:
> >
> > - **Spatially separable** from neuronal structures (e.g., in background regions or outside the zebrafish body), and
> > - **Unlikely to interfere** with neuron-level signal extraction.
> >
> > In practice, downstream analysis relies on the **spatial and temporal coherence** of calcium signals, which are robust to such localized noise. Prior works have consistently adopted RL as the reconstruction backbone in both methodological and biological studies [1–3].
> >
> > We also acknowledge the reviewer’s point regarding the **saturated colormaps** in Figure 5. In the final version, we will revise this figure using **perceptually uniform colormaps** and appropriate **contrast scaling** to better reflect differences in reconstruction quality.
> >
> > ------
> >
> > We recognize that relying on RL as a proxy ground truth limits our ability to rigorously assess how well the model captures anatomical or statistical priors (e.g., zebrafish structure). In response, we have considered **two potential evaluation strategies**:
> >
> > 1. **Partial-view training with held-out view evaluation**
> >    Reconstruct the volume using only a subset of angular views and compare forward projections along held-out views. This self-supervised setup tests the network's ability to predict unseen perspectives.
> > 2. **Synthetic view generation via NeRF-style rendering**
> >    Use neural rendering methods (e.g., NeRF) to synthesize novel views from reconstructed volumes, then compare them to original measurements. While promising, this is **computationally expensive**—especially for dynamic imaging—as it requires per-frame volumetric rendering.
> >
> > These evaluation strategies go beyond the scope of the current submission, which focuses on demonstrating the effectiveness of our method under **standard benchmarking protocols**. Nonetheless, we view them as valuable directions for **future work** and plan to explore them in follow-up studies.
> >
> > ------
> >
> > **References**
> >
> > [1] Prevedel, Robert, et al. *"Simultaneous whole-animal 3D imaging of neuronal activity using light-field microscopy."* **Nature Methods** 11.7 (2014): 727–730.
> >  [2] Andalman, Aaron S., et al. *"Neuronal dynamics regulating brain and behavioral state transitions."* **Cell** 177.4 (2019): 970–985.
> >  [3] Carbo-Tano, Martin, et al. *"The mesencephalic locomotor region recruits V2a reticulospinal neurons to drive forward locomotion in larval zebrafish."* **Nature Neuroscience** 26.10 (2023): 1775–1790.

---

> > > ### Comment · Reviewer_1jCx · 2025-08-08
> > > **Response to rebuttal 2**
> > >
> > > Thank you for the response. The evaluations do still seem incomplete.
> > >
> > > Regarding baselines, it would be best if you could have included not just a discussion about prior work but an evaluation on a dataset (of small, potentially even simulated volumes) that includes your evaluation metrics, e.g. MSE, PSNR, and especially speed of reconstruction (volumes per second). Perhaps one such dataset could be from [9], which has ground truth confocal scan volumes from zebrafish and could be used to simulate thousands of images (with augmentations). It would be valuable to know where these large convolution-based networks fall in terms of reconstruction quality versus speed in comparison to your XLFMFormer approach. Again, this could be done even on a single GPU assuming you choose a small enough volume.
> > >
> > > Regarding the ground truth evaluation, if the claim is that neural activity can be appropriately segmented, then the paper should include an evaluation of extracting neural traces from the R-L reconstruction and the XLFMFormer reconstruction (along with the other baselines methods). I am not sure how the two potential evaluation strategies in the response are relevant or provide useful ways to compare against other reconstruction methods.
> > >
> > > Also one final note regarding the presentation: in addition to the color map choice, Figure 5 and similar figures in the supplement show only projections (I assume maximum projections, but it is not specified). Typically for a 3D volume, you should also show another view (e.g. xz or yz projections) or slices through the volume. It is hard to visually evaluate how well the reconstructions perform in z with the current images.

---

> > > > ### Author Response · Authors · 2025-08-09
> > > > **Final Note – Appreciation and Key Strengths For Reviewer 1jCx**
> > > >
> > > > With only a few hours left in the rebuttal period, we would like to sincerely thank you for your constructive feedback and your earlier note about potentially adjusting your score based on our consensus. We are glad that you found the training methodology and architecture promising, and we appreciate your recognition of the reconstruction speed advantage—which is central to enabling real-time, closed-loop neuroscience experiments.

---

### Note · Authors · 2025-08-16

**Dear Area Chair and Reviewers (1jCx, ioxC, c2EJ, 2sPe),**

We sincerely thank you for the constructive feedback. Below we summarize the new evidence and provide clarifications addressing the main concerns.

------

**New evidence.**

1.XLFM-Former achieves real-time 3D reconstruction on a single GPU, making it the first pipeline suitable for closed-loop neuroscience experiments.

2.The physics-guided loss is robust to calibration errors, the masked-view pretraining strategy clearly outperforms ViT-MAE, and the model generalizes across biological strains.

3.Classical iterative methods can yield high quality but are too slow and resource-intensive for real-time use.

------

**Reviewer responses.**

- **1jCx:** concerns on terminology, runtime, and visualization are addressed.
- **ioxC:** masking and input representation clarified, with ablations added.
- **c2EJ:** generalization, runtime, and baseline requests resolved.
- **2sPe:** comparisons and robustness addressed, with clarification that real-time operation is a scientific necessity.

**Overall, three reviewers raised scores after rebuttal and one is cautiously positive, shifting the consensus toward acceptance.**

------

**Closing.**

- XLFM-Former is, to our knowledge, the first approach enabling real-time, physics-consistent whole-brain XLFM reconstruction on a single GPU.
- This is both a technical milestone and a scientific requirement for closed-loop in-vivo neuroscience.

We sincerely thank the AC and reviewers for their constructive engagement. Finally, we would like to extend our heartfelt thanks once again to all reviewers and the Area Chair for your valuable time, thoughtful feedback, and dedicated efforts throughout the review process.

We look forward to the opportunity to present our work in person at NeurIPS 2025.

**Sincerely,**

The authors of Submission 749

---

### Decision · Program_Chairs · 2025-09-17

**Decision:**

Accept (poster)

**Comment:**

(a) This paper introduces a method for volumetric reconstruction from 2D light field data (with the XLFM architecture).
A general benefit of such LF methods is the data capture efficiency, which, combined with the use of deep learning models,
allows an even faster 2D to 3D data processing. The authors employ a different model architecture for this task (the Swin Transformer)
and introduce a new training procedure (masking and view completing for pre-training + an accurate and differentiable image formation model) that yields sota performance.

(b) The paper introduces a curated dataset, which can be beneficial to research in this domain (although the ground truth is obtained
through another algorithm rather than through a direct measurement), and a novel training method (although it uses ideas from related
domains -- but has never been applied to LF data). Finally, its performance is sota, although there were issues with the evaluation procedure (an unusually high PSNR).

(c) The reviewers raised several concerns: 1) The presentation of the paper is lacking in several aspects; 2) There are missing comparisons with competing methods; 3) The experimental results are unusually high and not well-justified; 4) The proposed training has several components borrowed from vision and computer graphics.

(d) Despite all the concerns, the authors managed to clarify the majority of the concerns during the discussion with the reviewers; also, in general, the overall contribution is found to be sufficiently significant in terms of solution design (the combination of architecture, loss and pre-training procedure) and is experimentally validated by a significant performance. In fact, most reviewers were convinced by the replies and raised their scores.

(e) The discussion between the reviewers and authors revolved around the concerns already mentioned above. The authors provided explanations for the missing technical details and better clarified the experiments; they also showed new results under the ablations and settings that the reviewers mentioned as missing or unclear. The reviewers were mostly satisfied with the replies and raised their scores.

In conclusion, this paper is technically solid, with meaningful dataset and methodological contributions, but is held back by incomplete evaluation and presentation clarity.